# MATHMO: Automated Mathematical Modeling Through Adaptive Search

**Tennison Liu, Mihaela van der Schaar**
DAMTP, University of Cambridge
Cambridge, UK
`tl522@cam.ac.uk`

## Abstract

Mathematical modeling is the process of understanding and predicting complex real-world phenomena. Traditionally, it is a time-intensive effort reliant on deep human expertise and iterative refinement. Automating this intricate process, therefore, offers the potential to significantly accelerate discovery and broaden the application of mathematical modeling across diverse domains. Such automation, however, must address inherent challenges, including fundamental modeling uncertainty, balancing multiple conflicting objectives, and incorporating subjective qualities into assessing model utility. We approach this by conceptualizing mathematical modeling as a sequential decision-making problem under uncertainty. In response, we introduce MATHMO, a novel adaptive search method designed to automatically navigate the complex decisions in selecting mathematical frameworks, specifying model formulations, and defining algorithmic procedures. Specifically, MATHMO employs a principled bi-level search strategy—combining high-level exploration across diverse frameworks and local intra-framework model refinements—leveraging Large Language Models for exploration, surrogate evaluations, and incorporating subjective preferences into the automated process. We demonstrate MATHMO's efficacy on diverse real-world tasks, where it successfully discovers Pareto-efficient frontiers of models that balance varied objectives, including subjective criteria.

## 1 Introduction

Mathematical modeling is the art and science of translating complex real-world phenomena into precise mathematical language, allowing us to represent, understand, and predict complex situations. This capability is crucial in almost all aspects of life, from natural sciences and engineering to economics and social systems (Turing, 1990; Sugihara and May, 1990; Banwarth-Kuhn and Sindi, 2020). Indeed, the ability to abstract complex realities into mathematical models is often considered a key feature of intelligent civilization, enabling generalized problem-solving, knowledge transfer, and accumulation of scientific understanding over time (Simon, 2019).

The time-consuming and expertise-driven nature of modeling, coupled with its inherent complexities, makes its automation a highly appealing prospect. Automating this process could democratize access to powerful analytical models and tools, efficiently exploring trade-offs between multiple models, uncovering novel approaches or insights, and enhancing decision-making across diverse domains.

Several key characteristics distinguish the mathematical modeling process. Firstly, it is pursued in the face of fundamental *uncertainty*: the optimal framework or model specification is often unknown a priori, demanding iterative exploration involving building, testing, and refining models based on feedback (Jakeman et al., 2006). Secondly, modeling frequently contends with *multiple, often conflicting, objectives* (e.g., an optimization model's solution quality versus runtime (Chandrasekaran and Jordan, 2013)). Thus, the aim is typically not a single 'best' model, but a diverse frontier of models representing different trade-offs for the human modeler to investigate. Lastly, *subjective qualities* like interpretability and domain understanding further influence model utility beyond objective metrics (Dirac, 1963), making it crucial for them to be captured in the automated modeling process.

We conceptualize automated mathematical modeling as a sequential decision-making problem. Here, the modeler makes a series of choices: (1) the mathematical *framework* to employ, (2) the concrete

*model specification* including appropriate representations, parameterizations and assumptions, and (3) the appropriate *computational algorithms* to obtain the desired outputs. The uncertainty lies in not knowing which modeling decisions will yield useful models, thus requiring a principled approach of adaptive search that carefully balances exploration and exploitation given uncertainty.

In response, we introduce MATHMO, a novel method designed to automate key aspects of the mathematical modeling pipeline. At a high level, our system takes a modeling problem description and a set of objective functions, and discovers a set of models that represent efficient trade-offs among these objectives. The adaptive search procedure in MATHMO operates under a *bi-level structure*. The upper-level performs adaptive resource allocation across different mathematical frameworks, while the low-level module initiates a local search mechanism to explore each framework's model and algorithm space. Crucial to this process are *Large language models* (LLMs), which are employed to sample realizations from the search space and to perform surrogate evaluations to improve search efficiency. Additionally, they are used as models of subjective evaluations, thus incorporating subjective model preferences into the modeling pipeline. In each round, the generated model and its evaluations are observed, informing and adapting the next iteration of modeling decisions.

**Contributions.** The primary contributions of this work are threefold: **(1)** We formally define automated mathematical modeling, conceptualizing it as a sequential decision-making problem under uncertainty. **(2)** We present MATHMO, an adaptive search framework designed for this problem, capable of efficient exploration, balancing multiple objectives, and incorporating subjective modeling preferences. To the best of our knowledge, this is the first work to address this exciting problem area. **(3)** We demonstrate the efficacy of MATHMO on four diverse real-world modeling tasks (two prescriptive, two predictive), demonstrating its ability to discover Pareto-efficient frontiers of models.

## 2    PRELIMINARIES

### 2.1    FORMALISM

Mathematical modeling is primarily a *declarative* endeavor (Van Roy and Haridi, 2004). The core cognitive task involves translating a problem $p \in \mathcal{P}$ (potentially accompanied by a dataset $\mathcal{D}_p$) into a suitable formal representation, where the subsequent derivation of mathematical outputs is delegated to computational tools. We formalize this process of representation and derivation as a sequential decision-making problem under uncertainty (c.f. "Box's Loop" (Box, 1979)):

1. Selecting a high-level approach or *framework* $f \in \mathcal{F}$, where the chosen framework encodes foundational assumptions and provides access to specialized mathematical tools and techniques.
2. Specifying an exact *model* $m \in \mathcal{M}(f)$ within this framework, which is a concrete mathematical formulation representing the system under study.
3. Developing a computational or *algorithmic procedure* $a \in \mathcal{A}(f, m)$ to this model to derive the desired *mathematical output* $o = a \circ m(\mathcal{D}_p) \in \mathcal{O}$.
4. Evaluating the mathematical output or model characteristics to inform subsequent refinements.

This sequence of modeling decisions, denoted by $(f, m, a)$, defines a structured search space $\Omega = \{(f, m, a) \mid f \in \mathcal{F}(d), m \in \mathcal{M}(f), a \in \mathcal{A}(f, m)\}$. We evaluate each modeling outcome using a vector of $k \in \mathbb{N}$ *objective functions*, $\mathcal{J}(m, a) = [\mathcal{J}_1(m, a), \ldots, \mathcal{J}_k(m, a)]^T$. For generality, each objective $\mathcal{J}_i(\cdot, \cdot)$ can depend on both the mathematical outputs $o$ (e.g., solution optimality) and characteristics of the model-output pair (e.g., runtime). The goal is then to identify modeling decisions $(f, m, a) \in \Omega$ that address the following multi-objective optimization problem:

$$\text{minimize}_{(\cdot, m, a) \in \Omega}  \mathcal{J}(m, a) = [\mathcal{J}_1(m, a), \ldots, \mathcal{J}_k(m, a)]^T \tag{1}$$

As there typically does not exist a single model-algorithm pair that can minimize all objective functions simultaneously, we are more interested in finding the Pareto optimal models, i.e., models that cannot be improved in any of its objectives without degrading at least one. Mathematically, a Pareto optimal pair $(m, a)$ is non-dominated, where a pair $(m, a)$ is said to dominate another pair $(m', a')$ if $\forall\, i \in [k], \mathcal{J}_i(m, a) \leq \mathcal{J}_i(m', a')$ and $\exists\, i \in [k], \mathcal{J}_i(m, a) < \mathcal{J}_i(m', a')$ (Miettinen, 1999).

**Key challenges.** This problem definition emits several noteworthy challenges:

1. **Efficient exploration under uncertainty.** A defining hallmark of the mathematical modeling process is the fundamental uncertainty. It involves making sequential choices (framework, model,

Figure 1: **Overview of MATHMO.** Given a modeling problem desciption, MAMO employs a bi-level adaptive search strategy to identify a Pareto set of models presenting diverse trade-offs.

    algorithm) with uncertain outcomes in an interconnected and complex space. Efficient exploration, informed by prior beliefs and feedback, is thus crucial to navigating this search space.

2. **Fundamental trade-offs.** Modeling inherently involves balancing conflicting objectives, such as accuracy and interpretability. Different modeling frameworks (e.g., deep learning versus linear models) often embody fundamentally distinct trade-off frontiers, necessitating exploration across frameworks, beyond locally within a framework, to identify a set of Pareto efficient models.

3. **Subjective qualities of models.** Beyond objective metrics, subjective qualities like Occam's Razor, interpretability, or alignment with domain understanding are integral to a model's utility.[1] Although mathematical proxies for these elusive qualities exist (e.g., sparsity, minimum description length), they are typically framework-specific and not directly comparable.

## 2.2 RELATED WORKS

Our work builds upon and extends several lines of work:

**AutoML.** The field of AutoML aims to automate applying machine learning to real-world problems (Thornton et al., 2013). This broad endeavor encompasses hyperparameter optimization (Snoek et al., 2012; Li et al., 2018), neural architecture search (Zoph and Le, 2016; Pham et al., 2018), automated feature engineering (Khurana et al., 2016), and discovering loss functions (Real et al., 2020) or optimization algorithms (Chen et al., 2023b). Typically, these approaches predefine a custom search space (sometimes referred to as a domain-specific language), and apply search techniques such as Bayesian optimization, evolutionary algorithms, or bandit-based search (He et al., 2021). Our work shares this automation goal but differs significantly in scope, focusing on constructing mathematical models and their algorithmic procedures, navigating a more complex, open-ended search space than the narrower, pre-defined search spaces targeted by conventional AutoML.

The advent of LLMs has presented new opportunities for automated search problems. These large-scale pretrained models function as highly flexible generators, enabling search over problem spaces expressible through natural language (Brown et al., 2020) and overcoming bottlenecks in representation and search. They have been employed as zeroth-order optimizers over numerical spaces (Yang et al., 2024) (e.g., for hyperparameter optimization (Liu et al., 2024)). More strikingly, LLMs show remarkable efficacy in symbolic and combinatorial domains, in search spaces of reward functions (Ma et al., 2024), neural architectures (Chen et al., 2023a), symbolic expressions (Liu et al., 2025; Shojaee et al., 2025), and algorithms (Romera-Paredes et al., 2024).

**LLMs for mathematical formulations.** Our work is most closely related to emerging research using LLMs to generate mathematical formulations. Here, "formulation" means specifying a model within a predefined framework. Instances include formulating statistical models (Li et al., 2024), game-theoretic models (Mensfelt et al., 2024), dynamic systems (Holt et al., 2024), and convex optimization models (Ahmaditeshnizi et al., 2024). These typically assume a known modeling approach. Our work generalizes this line of research by not assuming a predefined framework. Instead, it automatically searches across diverse mathematical frameworks, removing a priori assumptions and enabling efficient exploration of diverse trade-offs offered by fundamentally different frameworks (e.g., metaheuristic optimization vs. convex optimization).

## 3 PROPOSED FRAMEWORK

The space of potential mathematical models for any given problem is inherently vast and complex. Navigating this nested, heterogeneous space with a flat exploration strategy is prone to inefficiencies.

---

[1]*"It is more important to have beauty in one's equations than to have them fit experiment"*—Paul Dirac

We propose an adaptive search framework that exploits structure in the modeling process to decompose this complexity, with the aim of improving efficiency (Dempe, 2002).

## 3.1 BI-LEVEL ADAPTIVE SEARCH

Our approach is informed by key observations about mathematical modeling. Firstly, the set of viable high-level modeling frameworks for a given problem is typically much smaller than the vast space of concrete models and algorithmic instantiations. This allows for more readily applicable priors on framework-level performance characteristics and suitability, for instance, incorporating coarse priors on the trade-off between performance and interpretability for deep learning versus mechanistic models. Secondly, performance variations between frameworks generally dominate those within them. The fundamental trade-offs offered by an exact method (e.g., integer programming) versus an approximate one (e.g., a metaheuristic) are typically more significant than those between different formulations under the same integer programming paradigm.

Based on these insights, we introduce a bi-level separation in the search process. The **upper-level search** explores different modeling frameworks, while the **lower-level search** focuses on discovering effective model formulations and solver/algorithm designs within that chosen framework. This bi-level separation, which mirrors the cognitive workflow often employed by human modelers, is expected to confer several advantages. Explicitly separating framework-level decisions allows for more effective exploration of model trade-offs. Different frameworks often populate distinct regions of this frontier, and the bi-level formulation helps systematically identify such trade-offs. Furthermore, modeling choices within a given framework tend to be structurally similar. This relative homogeneity means that feedback from one model instance provides a stronger signal for guiding improvements to related formulations within the same framework. For instance, insights from incorporating a logistic growth term into one dynamical system model are more directly transferable to refining another dynamical system's parameters than to designing the neural architecture of a deep model.

## 3.2 FORMAL DESCRIPTION

Mirroring the sequential nature of modeling decisions, our framework is formalized as an adaptive search process. The search proceeds in iterations indexed by $t = 1, 2, \ldots, T$. At each iteration $t$, decisions are informed by the history of previously explored models and their evaluated performance. Let $\mathcal{S}_{t-1} = \{(m_{t'}, a_{t'}, r_{t'})) | t' < t\}$ denote this history, where, for notational simplicity, we represent the objective value as $r_{t'} = \mathcal{J}(m_{t'}, a_{t'}) \in \mathbb{R}^k$. For the history within a particular framework $f$, we define $\mathcal{S}_{t-1}^f \subseteq \mathcal{S}_{t-1}$. The iterative process is decomposed into two nested levels:

**Upper-level problem.** At each iteration $t$, the upper-level decision involves selecting a modeling framework $f_t \in \mathcal{F}(p)$. This selection is guided by past performance across all explored frameworks:

$$f_t = \arg\max_{f \in \mathcal{F}(p)} \ \alpha(f; \mathcal{S}_{t-1}) \tag{2}$$

Here, $\alpha$ is a scalar *utility function* that estimates the potential value to explore framework $f$ at time $t$, given $\mathcal{S}_{t-1}$. This function, by quantifying preferences over frameworks, is crucial for managing the exploration-exploitation trade-off (Jones et al., 1998; Srinivas et al., 2010). For instance, $\alpha$ might prioritize frameworks that have recently yielded high-performing models (exploitation) or those less explored that could unveil novel regions of the Pareto frontier (exploration).

**Lower-level problem.** Once a framework $f_t$ is selected by the upper level, the lower-level problem focuses on identifying a new model pair $(m_t, a_t)$ within the space of $\mathcal{M}(f_t) \times \mathcal{A}(f_t, m)$. This local exploration leverages the historical performance of evaluated pairs within the framework:

$$(m_t, a_t) = \arg\max_{(m,a) \in \mathcal{M} \times \mathcal{A}} \beta_{f_t}(m, a; \mathcal{S}_{t-1}^{f_t}) \tag{3}$$

Here, $\beta_{f_t}$ is a framework-specific utility function that learns from evaluations of past pairs $\mathcal{S}_{t-1}^{f_t}$ explored within framework $f_t$ to estimate the potential value of a candidate pair $(m, a)$. Together, Equations (2) and (3) define an iterative loop that systematically explores the model space, leveraging the bi-level structure to balance broad exploration across frameworks with focused refinement within them. Once $(f_t, m_t, a_t)$ are obtained, they are then evaluated to obtain $r_t$, and added to the history.

# 4  MATHMO: AUTOMATED MATHEMATICAL MODELING WITH LLMS

In what follows, we describe the Automated Mathematical Modeler (MATHMO), our specific implementation of the adaptive search framework (for an algorithmic overview, see Section C.1).

## 4.1  LLM SEARCH OPERATORS

Conventional automated search methods typically necessitate a clearly defined search space and a formal solution representation, often through a domain-specific language (DSL) (Hutter et al., 2011; 2019). The domain of general mathematical modeling, however, presents a significant challenge: the space of potential mathematical objects is extraordinarily vast and diverse, rendering the a priori definition of a comprehensive DSL or a fully structured search space practically infeasible.

To navigate this expansive and ill-defined landscape, MATHMO leverages Large Language Models (LLMs) as core search operators. Modern LLMs, pre-trained on massive corpora of text and code, encapsulate extensive knowledge across numerous domains, including mathematics, scientific literature, and programming (Brown et al., 2020; Kaplan et al., 2020). This pre-training endows them with strong implicit domain priors, which can be harnessed to guide the exploration of plausible and potentially effective mathematical modeling choices, moving beyond the limitations of rigidly defined search spaces. In MATHMO, LLMs fulfill two crucial roles in the bi-level search process:

1. **Generative samplers.** LLMs are employed to sample from the space of frameworks, as well as model and algorithmic specifications. In our implementation, specific models and algorithms are represented as executable Python code, while high-level frameworks are represented as textual descriptions. Conditioned on the problem description $p$, LLMs are prompted to sample suitable modeling frameworks, denoted as $f \sim p_\theta(\cdot \mid p)$. For a selected framework $f$, the LLM generates specific model and algorithmic instantiations, i.e., $(m, a) \sim p_\phi(\cdot, \cdot \mid p, f, \mathcal{S}^f)$, conditioned on the problem, framework, and past examples belonging to that framework.[2]

2. **Surrogate models.** LLMs also function as surrogate models to estimate the objective value of proposed model-algorithm pairs and inform the utility functions to guide search. Specifically, $\hat{r} \sim p_{\text{SM}}((m, a) \mid p, f, \mathcal{S}^f)$, where the subscript SM is employed to denote the surrogate model. Additionally, we also use LLMs as Surrogate Models Of Subjective Evaluations (MOSE). This surrogate, i.e., $\hat{r} = p_{\text{MOSE}}((m, a) \mid p)$, predicts subjective quality scores (e.g., human-perceived interpretability) based on model representation and output, which are integrated into the overall evaluation, allowing subjective inductive biases to be incorporated into search.

The operation of LLMs in these roles relies on specific prompts. The details are provided in Section C, but they follow a standard "skeleton" structure, incorporating the problem description $p$, current context (e.g., selected framework $f$, and history $\mathcal{S}$), and the specific task for the LLM.

## 4.2  UPPER-LEVEL PROBLEM: FRAMEWORK SELECTION

The upper-level problem addresses the decision of which modeling framework $f_t$ to commit to for an additional step of exploration. In MATHMO, we employ the Pareto Upper Confidence Bound (Pareto-UCB) strategy to realize the framework selection utility $\alpha(f, \mathcal{S}^f_{t-1})$ (Equation (2)). This method navigates the inherent multi-objective trade-offs by identifying a frontier of frameworks that are optimistically non-dominated, thus balancing the need to explore new avenues with exploiting proven ones (Drugan and Nowe, 2013; Xu and Klabjan, 2023).

At the beginning of search ($t = 0$), an initial set of candidate frameworks is proposed by employing the LLM-based sampler $f \sim p_\theta(\cdot \mid p)$. Each framework is initialized with an optimistic estimate of its potential, by setting an infinite upper confidence bound (UCB) value. This ensures that each framework is selected for at least one initial exploration cycle, providing data for subsequent, more informed decisions. Specifically, for each framework $f$, the historical performance vectors $\{r_{t'} \mid (m_{t'}, a_{t'}, r_{t'}) \in \mathcal{S}^f_{t-1}\}$ are used to estimate the empirical mean $\hat{\mu}_f \in \mathbb{R}^k$ and variance $\hat{\sigma}^2_f \in \mathbb{R}^k$. This is then used to calculate the UCB vector $\text{UCB}_f \in \mathbb{R}^k$, where each component is computed using

---

[2]Here $\phi, \theta$ are used to denote the prompts that module that sampling distribution (Sumers et al., 2023).

a formula that considers both the estimated mean and its uncertainty, encouraging exploration:

$$\text{UCB}_{f,j} = \hat{u}_{f,j} + c\sqrt{\frac{\hat{\sigma}_{f,j}^2 \ln(N_{t-1})}{N_{f,t-1}}} + d\sqrt{\frac{\ln(N_{t-1})}{N_{f,t-1}}} \tag{4}$$

where $\hat{\mu}_{f,j}$ and $\hat{\sigma}_{f,j}^2$ are the estimated mean and variance of objective $j$ of framework $f$, $N_{f,t-1}$ is the number of times framework $f$ has been evaluated, and $N_{t-1}$ is the total number of exploration steps (across all frameworks). $c$ and $d$ are hyperparameters that control the exploration bonus.

The set of $\text{UCB}_f$ vectors, one for each framework, is then used to identify a subset of the promising frameworks. A framework $f$ is considered part of the Pareto optimal set, if its $\text{UCB}_f$ is non-dominated (i.e., $\text{UCB}_{f,j} \geq \text{UCB}_{f',j} \, \forall j \in [k]$ and $\exists j \in [k] : \text{UCB}_{f,j} > \text{UCB}_{f',j} \, \forall f'$, assuming maximization). From this Pareto-UCB set, one framework is randomly selected to be explored in the next iteration.

## 4.3  LOWER-LEVEL PROBLEM: LOCAL EXPLORATION

Once the upper-level process selects a framework $f_t$, the lower-level is concerned with performing local exploration within the chosen framework to identify promising $(m_t, a_t)$ for subsequent evaluation. MATHMO achieves this through a three-stage process that involves first sampling candidate pairs, performing surrogate evaluations, and finally selecting the most promising one based on these predictions, in a process akin to Bayesian Optimization (Snoek et al., 2012; Liu et al., 2024).

First, a set of diverse candidate model-algorithm pairs are sampled, which we denote as $\tilde{\mathcal{S}}^f = \{(\tilde{m}^{(i)}, \tilde{a}^{(i)}) \mid i \in [l]\}$, where each $(\tilde{m}^{(i)}, \tilde{a}^{(i)}) \sim p_\phi(\cdot, \cdot \mid p, f_t, \mathcal{S}_{t-1}^{f_t})$. For each sampled candidate pair, we then estimate its $k$-dimensional objective vector using LLMs as a surrogate model: $\hat{r}^{(i)} = p_{\text{SM}}(\tilde{m}^{(i)}, \tilde{a}^{(i)} \mid p, f_t, \mathcal{S}_{t-1}^{f_t})$, which provides a low-cost prediction of how each candidate might perform if fully evaluated, with each of the $k$ objectives estimated independently.

Given the predicted objectives, MATHMO performs selection based on maximizing the estimated hypervolume (Guerreiro et al., 2020). Intuitively, a higher estimated hypervolume means the pair is more likely to dominate a larger portion of the $k$-dimensional objective space, relative to a reference point. Formally, $\text{HV}(\tilde{m}^{(i)}, \tilde{a}^{(i)}; r_{\text{ref}})$, where $r_{\text{ref}} \in \mathbb{R}^k$ is the reference point. To account for potentially different scales of the $k$ objectives, the individual objective values are normalized to $[0, 1]$ before calculation, and $r_{\text{ref}}$ is set as $1_k$. The pair $(m_t, a_t)$ is then selected as the candidate that yields the largest hypervolume: $(m_t, a_t) = \arg\max_{(\tilde{m}, \tilde{a}) \in \tilde{\mathcal{S}}} \text{HV}(\tilde{m}, \tilde{a}; r_{\text{ref}})$.

## 4.4  MOSE: SURROGATE MODEL OF SUBJECTIVE EVALUATIONS

Mathematical modeling is not solely guided by objective performance metrics; subjective qualities, such as interpretability and alignment with domain knowledge, are crucial utility considerations for human modelers. Indeed, models generally reflect how we conceptualize and understand complex situations. Incorporating these aspects into an automated search is thus vital to ensure the generated models are useful, amenable to further analysis, and capable of communicating valuable insights. While framework-specific metrics like complexity or sparsity penalties can promote such qualities (e.g., in symbolic or linear regression), they are often not transferable across different paradigms, limiting their utility when the goal is to compare diverse models spanning multiple frameworks.

To address this, we introduce MOSE as a generalized, cross-framework mechanism for integrating subjective criteria into model evaluation. We acknowledge that subjective qualities can be highly observer-dependent; MOSE therefore aims not to capture perfect objectivity but to provide consistent surrogate approximations. This approach is based on the observation that subjective qualities are often more reliably expressed through comparative evaluations than absolute scores—an insight also exploited in Reinforcement Learning from Human Feedback (Bradley and Terry, 1952; Christiano et al., 2017). Furthermore, we leverage the ability of advanced LLMs to simulate human judgments, enabling scalable preference elicitation without costly annotations, a technique proven effective in domains like alignment and diversity search (Bai et al., 2022; Bradley et al., 2024).

To ensure comparability of subjective evaluations across diverse discovered models, MOSE employs a predefined reference set of models, denoted as $\mathcal{M}_{\text{ref}}$. This set, generated at the start of the search, remains fixed as a consistent frame of reference. When evaluating a new model $m_t$ on

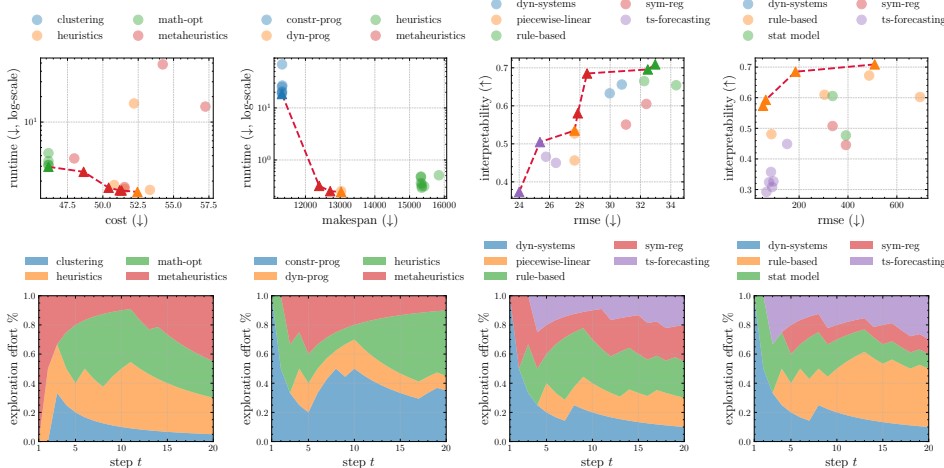

Figure 2: **Pareto fronts and adaptive exploration. (Top)** Pareto fronts of models produced by MATHMO on TSP, JSS, Ecology, and Epidemiology tasks. **(Bottom)** Corresponding cumulative % of exploration effort allocated to each modeling framework throughout the search process.

a specific subjective quality (e.g., interpretability), MOSE performs pairwise comparisons against each reference model. Specifically, it predicts '1' if the evaluated model is more preferable, and '0' otherwise. The probabilities associated with these predictions are then averaged to obtain a score: $\hat{r} = \frac{1}{|\mathcal{M}_{\text{ref}}|} \sum_i p_{\text{MOSE}}(m_t \succ m_i \mid p)$. By averaging over a fixed reference set, this approach yields a consistent score in $[0, 1]$, and mitigates potential sensitivities to the choice of reference baseline.

## 5 EXPERIMENTS

**Research questions.** In this section, we present an empirical evaluation of MATHMO. Our experiments are designed to investigate the following research questions:

1. How does MATHMO perform on diverse, real-world tasks, particularly its ability to discover a range of models that effectively navigate different trade-offs (Section 5.1).
2. What are the contributions of specific algorithmic design decisions within MATHMO to search performance, as analyzed through controlled ablation studies (Section 5.2).
3. How effectively does MOSE capture and integrate subjective modeling preferences, such as interpretability, into the automated modeling process (Section 5.3).

**Problems.** To address these questions, we employ four distinct modeling problems, two prescriptive and two predictive, each presenting unique challenges and trade-offs: **Job Shop Scheduling (JSS):** This problem involves optimally scheduling jobs on machines, subject to precedence/resource constraints. We investigate the trade-off between **makespan** (total completion time) and **runtime**, using 10 instances of varying complexities (50-300 ops). **Traveling Salesman (TSP):** This NP-hard problem seeks the shortest route visiting each location exactly once. We examine the trade-off between **route cost** (total tour length) and **runtime** (time to find a tour), utilizing 10 instances of diverse complexities (30-50 locations). **Ecology:** This involves understanding and predicting population dynamics in ecological environments. We focus on the trade-off between **predictive performance** (measured by RMSE on unseen data) and **interpretability** (assessed by MOSE). We use a real-world two-species dataset. **Epidemiology:** The goal is to understand and simulate the spread of infectious diseases. We investigate the trade-off between **predictive performance** (RMSE) and **interpretability**, employing a real-world COVID-19 dataset from Italy.

In these problems, modeling trade-offs are crucial. For instance, operations managers might use slower, more optimal models for long-term planning (JSS, TSP), yet rapid, near-optimal solutions are invaluable for dynamic rescheduling or handling disruptions. Similarly, while ecologists and epidemiologists need high predictive accuracy, model interpretability is vital for gaining scientific insights into underlying mechanisms (e.g., population dynamics, disease transmission) and informing effective interventions like conservation strategies or public health policies.

For all experiments, we run MATHMO for 20 iterations. This process starts with proposing an initial set of 5 frameworks, and each model evaluation is subject to a time limit of 300 seconds. For MOSE,

Table 1: **Ablation study.** Hypervolume performance and relative improvements achieved by MATHMO.

| Ablations | TSP | JSS | Ecology | Epidemiology | % Improvement |
|---|---|---|---|---|---|
| MATHMO | 0.998 | 0.994 | 0.992 | 0.967 | - |
| MATHMO_RAN | 0.972 | 0.948 | 0.992 | 0.939 | 2.552% |
| MATHMO_FLAT | 0.987 | 0.945 | 1.000 | 0.792 | 5.848% |
| MATHMO_NAIVE | 0.977 | 0.973 | 0.894 | 0.713 | 10.096% |

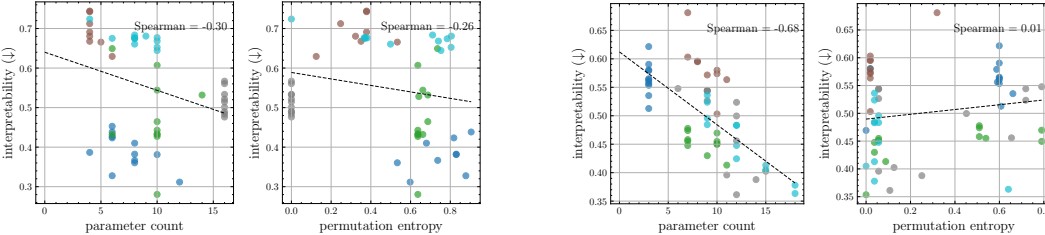

Figure 3: **Evaluation of MOSE.** Correlation analysis of MOSE interpretability scores against structural and functional complexity metrics on ecology task **(Left)** and epidemiology task **(Right)**.

a reference set comprising 3 models is employed. We use gpt-4o-2024-05-13 as the LLM. Additional details on datasets/experimental setup are provided in Section D.

**Additional results.** In the interest of space, we provide extended experiments and analyses in Section B. We first establish *generality* beyond our core benchmarks by evaluating MATHMO on two large-scale medical risk prediction domains, NHANES and SEER (Section B.1), which involve expensive model training and a distinct multi-objective trade-off between discriminative performance (AUROC ↑) and calibration (ECE ↓). We next benchmark against strong baselines, including mature domain-specialized solvers and AutoML-style symbolic regression systems (Section B.2), as well as recent LLM-based heuristic design methods (Section B.3).

We then assess robustness and internal dynamics through framework removal and inter-run consistency analyses (Section B.4), comparisons of discovered Pareto fronts and hypervolume trajectories under ablations, and targeted studies of upper-level framework selection and lower-level surrogate-guided exploration (Sections B.6 and B.7). Finally, we provide additional validation of MOSE (Section B.8), including sensitivity analyses and expert human alignment studies.

## 5.1 PERFORMANCE ON DIVERSE TASKS

The performance of MATHMO across the four modeling tasks is visualized in Figure 2. Panel **(Top)** of the figure displays the Pareto fronts of discovered models for each task, while panel **(Bottom)** illustrates the corresponding framework exploration dynamics (cumulative effort allocation).

**Frameworks dominate trade-offs.** A consistent finding is that different modeling frameworks tend to excel in different regions of the Pareto frontier, underscoring the importance of framework selection. For the JSS and TSP tasks, exact methods such as mathematical optimization and constraint programming yield solutions closer to optimality but incur significantly higher computational runtimes. Conversely, metaheuristics and custom heuristics provide solutions with much faster runtimes, with a trade-off in solution quality. This pattern extends to the Ecology and Epidemiology tasks. Here, time-series forecasting methods like vector autoregression achieve strong predictive performance but are assessed by MOSE as less interpretable. In contrast, frameworks such as dynamical systems (e.g., compartmental models) are considered more interpretable, though they exhibit higher RMSE.

**Effective intra-framework refinement.** MATHMO demonstrates effectively exploration within frameworks to refine solutions along the Pareto front, for instance, refining solver heuristics on JSS/TSP and redesigning simulated annealing-based metaheuristics. Similarly, for the Ecology and Epidemiology tasks, when exploring within the dynamical systems framework, MATHMO proposes variations in model specification (e.g., logistic growth terms, interaction terms) to improve predictive accuracy.

**Adaptive exploration behavior.** The framework exploration dynamics reveal adaptive search behavior, characteristic of UCB-family strategies. On JSS, TSP, and Epidemiology, we observe a clear pattern of initial broad exploration followed by focused exploitation on frameworks that contribute to the Pareto-UCB frontier. Interestingly, on the Ecology task, the exploration allocation remains more evenly distributed among several distinct frameworks (time-series, dynamical systems, rule-

based models), which is consistent with the observation that these frameworks contribute unique, non-dominated solutions to different regions of the Pareto frontier.

## 5.2 CONTROLLED ANALYSIS OF ALGORITHMIC COMPONENTS

Next, we turn to understanding the contribution of the design decisions of MATHMO. For these purposes, we evaluate three ablations: (1) MATHMO$_{RAN}$, where the Pareto-UCB framework selection strategy is replaced with random selection. (2) MATHMO$_{FLAT}$, which collapses the bi-level search into a flat search space, applying a globalized version of the local exploration mechanism. (3) MATHMO$_{NAIVE}$, which omits the surrogate-guided local exploration, relying solely on direct sampling.

The comparative performance of MATHMO and its ablated versions, as detailed by hypervolume in Table 1, reveals several key insights. The complete MATHMO generally achieves the best hypervolume, with one notable exception on the Ecology problem, where MATHMO$_{FLAT}$ (without the bi-level structure) found Pareto-dominant solutions by concentrating its search on dynamical systems. Employing random framework selection (MATHMO$_{RAN}$) resulted in an average hypervolume decrease of $2.5\%$, underscoring the value of adaptive exploration at the framework level. The removal of bi-level search structure had a more pronounced negative impact on model diversity; manual examination indicated this ablation tended to overconcentrate (allocating $95\%$ of exploration effort to metaheuristics on JSP/TSP and $80\%$ on ecology). MATHMO$_{NAIVE}$, which relies on repeated LLM sampling, exhibits the poorest performance overall. However, it interestingly achieved a better hypervolume than MATHMO$_{FLAT}$ on the JSS task, suggesting that even naive, broad sampling by the LLM could sometimes provide better coverage of the Pareto frontier than an overly myopic flat search.

## 5.3 SUBJECTIVE QUALITY EVALUATIONS

In this final experimental section, we assess how well MOSE captures aspects of model interpretability. Quantifying interpretability directly is challenging in the general sense, and the broader utility MOSE lies in its generalizability across different frameworks and problems. Fortunately, for the time-series problems in our benchmark, we can analyze MOSE's scores against two commonly used proxy complexity metrics: *(1)* structural complexity measured by the number of free parameters (fewer parameters often correlate with more understandable models), and *(2)* functional complexity assessed using permutation entropy of the model's predicted time-series. Permutation entropy quantifies the regularity and predictability of a time-series, with lower entropy suggesting simpler, more regular dynamics, while higher entropy indicates more chaotic patterns (Bandt and Pompe, 2002).

Our analysis, with detailed correlations presented in Figure 3, yields several insights into MOSE's behavior. Firstly, MOSE scores tend to cluster by modeling framework (e.g., dynamical systems and rule-based models consistently receive higher interpretability compared to autoregressive forecasting). The scores exhibit a statistically significant negative correlation with structural complexity on both tasks: Spearman correlation $\rho = -0.678$ ($p = 1.31 \times 10^{-8}$) for Epidemiology and $\rho = -0.298$ ($p = 0.0318$) for Ecology. The relationship with functional complexity appears more context-dependent. A significant negative correlation is observed on Ecology ($\rho = -0.261$, $p = 0.0313$), but almost no correlation is identified on Epidemiology. It is important to note that these proxy metrics (parameter count and permutation entropy) are themselves indirect measures of interpretability and are primarily employed here for analytical validation within these specific time-series contexts.

## 6 DISCUSSIONS

Automated mathematical modeling represents an exciting frontier in applying artificial intelligence to complex social, scientific, and engineering problems. In this work, we advance this frontier by characterizing the process as a sequential decision-making problem under uncertainty. We proposed a novel adaptive search framework designed to navigate the complex modeling space, featuring mechanisms capable of efficient exploration, balancing multiple objectives, and incorporating subjective preferences. Our concrete instantiation, MATHMO, demonstrates the potential of leveraging LLMs as versatile search operators within a structured bi-level search architecture. Empirical results across real-world modeling tasks underscore MATHMO's efficacy in discovering diverse frontiers of models, providing decision-makers with a rich set of alternatives offering different trade-offs.

**Future directions.** This research opens numerous avenues for future exploration and enhancement. Currently, the set of modeling frameworks is sampled and fixed at the beginning of the search. Future work could explore dynamic framework generation, enabling the system to discover or construct new modeling paradigms based on accumulated insights, rather than being confined to an initial set. While designed with multi-objective scenarios in mind, the framework can naturally be applied to single-objective modeling tasks, and further investigation could optimize its performance for such cases. Beyond `MOSE` for subjective preferences, developing richer interactive mechanisms for human experts to influence the search could significantly enhance outcomes. Lastly, the framework's general nature permits the development of more specialized utility functions and optimizations of LLMs as samplers or surrogate models, potentially yielding further performance gains. We hope that `MATHMO` and the proposed framework lay a useful foundation for future advancements in this domain.

ETHICS AND REPRODUCIBILITY STATEMENTS

**Ethics.** This work evaluates public benchmarks and de-identified real-world datasets (e.g., NHANES and SEER) used in accordance with their respective data usage policies. No identifiable personal data are accessed or stored. When employing external LLM services, we follow provider guidelines to ensure that submitted data are not retained or used for human review. Our study focuses on methodological development and does not involve real-world deployment or decision-making.

**Reproducibility.** Experimental investigations are described in Section 5 with further details of the method, experimental setup, and datasets included in Sections B to D. We provide the code to reproduce our results at https://github.com/tennisonliu/MATHMO.

ACKNOWLEDGMENTS

We thank the anonymous ICLR reviewers, members of the van der Schaar lab, and Andrew Rashbass for many insightful comments and suggestions. TL would like to thank AstraZeneca for their sponsorship and support. This work was supported by Microsoft's Accelerate Foundation Models Academic Research initiative.

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

# Appendix

## Table of Contents

## A    ADDITIONAL DISCUSSIONS

In the following section of the appendix, we offer additional discussion to motivate the need for automated mathematical modeling (`MATHMO`), reflecting the multi-objective trade-offs, variety of frameworks, and role of subjective criteria inherent in real-world modeling. We then outline potential enhancements to key components of the adaptive search framework.

### A.1    REAL-WORLD MODELING: UNCERTAINTY, TRADE-OFFS, AND SUBJECTIVE CRITERIA

Table 2: **Overview of problems.** Illustrating multi-objective trade-offs, modeling diversity, uncertainty, and subjective considerations.

| Problem | Objectives | Modeling Approaches | Uncertainty in Modeling | Subjective Criteria |
|---|---|---|---|---|
| **TSP** | ▷ Journey cost, ▷ Runtime | △ ILP, △ Metaheuristics, △ Heuristics | Instance size and structure affect performance; solver behavior is unpredictable | - |
| **Job Shop Scheduling** | ▷ Makespan, ▷ Runtime | △ Constraint programming, △ Metaheuristics, △ Heuristics | Solution quality and runtime vary with problem characteristics; method choice is non-obvious | - |
| **Ecology** | ▷ Predictive accuracy, ▷ Interpretability | △ Differential equations, △ ARIMA, △ Graphical models | Complex, noisy dynamics; uncertain model structure; missing or sparse data | Alignment with ecological knowledge and interpretability |
| **Epidemiology** | ▷ Forecast accuracy, ▷ Interpretability | △ Compartmental models, △ Statistical models, △ Time-series models | Highly sensitive to data quality and regime changes; difficult to validate assumptions | Interpretability for public health communication |
| **Medical Diagnosis** | ▷ Accuracy, ▷ Explainability, ▷ Uncertainty calibration | △ Deep learning, △ Rule-based systems, △ Probabilistic models | Variation in populations, equipment, and labeling; generalization is uncertain | Clinical trust and alignment with expert reasoning |
| **Portfolio Optimization** | ▷ Expected return, ▷ Risk, ▷ Robustness | △ Classical finance models, △ ML models (e.g., RNNs) | Financial time series are non-stationary; market conditions shift unpredictably | Transparency, explainability, and risk alignment |
| **Drug Response Modeling** | ▷ Predictive accuracy, ▷ Biological interpretability, ▷ Safety | △ Mechanistic models, △ Multi-omics ML, △ Hybrid causal models | High patient variability; limited, expensive data; strong prior assumptions needed | Regulatory approval and clinical interpretability |

This work addresses a relatively underexplored area: the automation of mathematical modeling under real-world constraints. Our problem formulation and methodological framework are motivated by three core challenges commonly encountered in practice: **(1)** uncertainty in modeling decisions—such as which frameworks or assumptions are most appropriate; **(2)** multi-objective trade-offs between competing criteria like accuracy, runtime, and interpretability; and **(3)** subjective, human-centric considerations that are difficult to mathematically formalize but critical to real-world adoption.

Our analysis focuses on domains such as job-shop scheduling, vehicle routing, ecology, and epidemiology, each of which presents unique modeling challenges and trade-offs, such as optimality versus runtime or predictive accuracy versus interpretability. However, these issues are far from domain-specific. In the following discussion, we illustrate how similar concerns arise across a wide range of modeling scenarios, further underscoring the importance of a general-purpose, flexible approach to automated modeling. An overview of the discussion is summarized in Table 2.

**Medical diagnosis.** The task of predicting diseases or conditions based on clinical data such as medical images, lab results, or patient history.

1. **Trade-offs:** ▷ Diagnostic accuracy (essential for minimizing missed or incorrect diagnoses) *vs.* ▷ explainability (clinicians need to understand model reasoning to trust and act on predictions) *vs.* ▷ uncertainty calibration (important for risk-aware decision-making, especially in borderline cases).
2. **Possible modeling frameworks:** △ Deep learning (highly accurate, data-intensive, low interpretability); △ rule-based expert systems (e.g., risk scores; interpretable but often underperform, and lack uncertainty handling); △ probabilistic models (capture uncertainty but require strong assumptions and are harder to scale).
3. **Subjective criteria:** Clinicians value interpretability and alignment with domain knowledge—understanding why a model makes a diagnosis is as important as the prediction itself.
4. **Uncertainty in modeling:** Variation in patient populations, medical instrumentation (e.g., imaging devices), and comorbidities makes it unclear which modeling assumptions will generalize. Ground truth labels may also be noisy or inconsistent across annotators.

**Financial portfolio optimization.** The process of allocating assets to maximize returns while managing risk in dynamic market environments.

1. **Trade-offs:** Expected return (central to investor objectives) *vs.* ▷ risk (higher returns generally entail higher volatility); ▷ predictive performance (important for exploiting market inefficiencies) *vs.* ▷ robustness (models may overfit to past data and fail under new market regimes).
2. **Possible modeling frameworks:** △ Classical financial models (e.g., Markowitz, Black-Litterman; principled but sensitive to input estimation errors); △ machine learning models (e.g., RNNs; flexible, can learn patterns, but require large, clean datasets and may overfit or lack robustness).
3. **Uncertainty in modeling:** Market dynamics are non-stationary and hard to forecast. Expected returns, volatilities, and correlations shift over time, making it unclear which models or assumptions will remain valid. A model that performs well in one regime/time horizon may fail in another.

**Drug response modeling.** Predicting how individual patients will respond to a given drug, often in the context of personalized medicine or drug development.

1. **Trade-offs:** ▷ Predictive accuracy (critical for identifying effective treatments) *vs.* ▷ biological interpretability (important for understanding mechanisms and gaining trust); ▷ short-term efficacy (desired for immediate outcomes) *vs.* long-term safety (essential for regulatory approval and patient well-being).
2. **Possible modeling frameworks:** △ Mechanistic models (e.g., PK/PD; grounded in biology, interpretable, but slow and parameter-sensitive); △ multi-omics ML models (data-driven and expressive, but opaque and difficult to validate); △ hybrid models (e.g., causal or semi-mechanistic; combine strengths, but sensitive to misspecification).
3. **Subjective criteria:** Interpretability and biological plausibility are crucial for clinical trust and regulatory acceptance.
4. **Uncertainty in modeling:** High variability across patients (e.g., in genetics or metabolism) complicates model generalization. Data is often limited, expensive to obtain, and ethically constrained.

In all these examples, the goal is not to identify a single best model, but rather to present the human user with a diverse set of viable modeling options—each representing different trade-offs across relevant objectives such as accuracy, interpretability, robustness, and runtime. This is crucial because the optimal modeling choice often depends on context-specific constraints, user preferences, and shifting priorities. Framing this as a multi-objective optimization problem allows us to systematically explore the space of trade-offs and approximate the Pareto frontier, enabling users to make informed decisions based on their own criteria and operational needs.

## A.2 ENHANCING COMPONENTS OF THE ADAPTIVE SEARCH FRAMEWORK

Our work presents a general framework for approaching automated mathematical modeling. Specifically, it decomposes the automated modeling process into a bi-level search: the upper level selects among modeling frameworks while managing exploration–exploitation trade-offs, and the lower level performs local search within each framework to identify high-performing model-algorithm pairs. While our specific instantiation of this framework demonstrates effectiveness, we outline key areas where individual components of this framework could be improved to enhance performance, flexibility, and robustness.

### A.2.1 UPPER-LEVEL UTILITY FUNCTION

We use Pareto-UCB to guide framework selection by approximating the Pareto frontier across multiple objectives. This balances exploration and exploitation in a principled way. However:

1. **Non-stationarity.** Pareto-UCB assumes a stationary reward distribution, which is perhaps untenable in our setting—modeling performance is expected to improve over time due to ongoing low-level exploration. This results in a shifting reward distribution and suggests that a dynamic or non-stationary approach may be more appropriate.
2. **Independence assumption.** Frameworks are currently treated as independent in the selection problem, but in reality, their performance is often correlated (e.g., if model-based control methods perform well, similar model-based RL might too). Ideally, inter-framework correlations and structure are captured somehow, although modeling or learning this correlation a-priori remains challenging.

3. **Alternative utility functions.** Optimism-based acquisition strategies like Pareto-UCB presents a strong initial approach, but Bayesian acquisition functions (e.g., entropy search) may better capture uncertainty and trade-offs in this multi-objective setting and are worth investigating in future work.

### A.2.2 LOWER-LEVEL UTILITY FUNCTION

Our lower-level utility function relies on an LLM-based surrogate to estimate the objective performance of candidate models, using expected hypervolume improvement as the acquisition criterion.

1. **Surrogate models.** Developing a surrogate model on this non-conventional search space (i.e., space of models expressed in natural language) is very challenging. While our work uses an LLM-based surrogate model and demonstrated improved search efficiency as a result, this is a fruitful area with open challenges in calibration and generalization.
2. **Alternative search strategies.** Other local search methods, such as evolutionary algorithms or learned reinforcement learning-based search policies, could offer more robust and generalizable exploration under uncertainty.
3. **Framework-specific search policies.** More interestingly, specialized framework-specific exploration strategies that exploit the structure unique to each framework could greatly improve efficiency. For instance, leveraging the hierarchical structures in mathematical programming (Astorga et al., 2024) or neural architecture search for deep learning models (Zoph and Le, 2016). However, this requires engineering these framework-specific strategies beforehand.

### A.2.3 LLM GENERATIVE SAMPLERS

LLMs are crucial for open-ended exploration in the space of frameworks, models, and algorithms.

1. **Finetuned search operators.** In our work we used general purpose LLMs as samplers, although there is no principled reason this could not be improved with specific finetuned search operators.
2. **Dynamic sampling.** When it comes to framework sampling, our current approach samples and fixes a set of frameworks at the beginning of search. Ideally, this could be improved with a dynamic approach, which is less constraining, enables more efficient exploration budget allocation, and encourages the emergence of hybrid or novel modeling paradigms.

### A.2.4 LLM SURROGATE MODELS

LLMs, which operate directly on natural language inputs, offer a powerful alternative to traditional surrogate models like Gaussian Processes that are limited to well-defined numerical feature spaces. This flexibility enables surrogate modeling over open-ended model descriptions, expanding the expressive range of the search process. However, LLM-based surrogates also come with notable challenges. Their uncertainty estimates tend to be poorly calibrated (Ling et al., 2024), and their outputs can be highly sensitive to prompt design and formatting (Xiang et al., 2024; Hwang et al., 2025). Addressing these issues—through techniques such as prompt ensembling, temperature scaling, and uncertainty-aware decoding—remains an important direction for improving both reliability and performance in LLM-based surrogate evaluation.

### A.2.5 MOSE: MODEL OF SUBJECTIVE EVALUATION

A general-purpose, cross-framework model of subjective evaluation introduces significant potential for modeling human-like preferences in automated modeling pipelines. Our initial results showed that MOSE scores are meaningfully correlated with structural and functional properties of candidate models, suggesting alignment with domain-relevant heuristics. Nonetheless, the model remains in an early stage and would benefit from further validation and tuning across diverse problem types. Encouragingly, similar challenges have been tackled in reinforcement learning from human feedback (RLHF), resulting in reliable and robust preference models (Christiano et al., 2017). In our setting, MOSE could be further improved using paired preference labels gathered from domain experts, enabling data-efficient finetuning and deeper alignment with subjective modeling criteria.

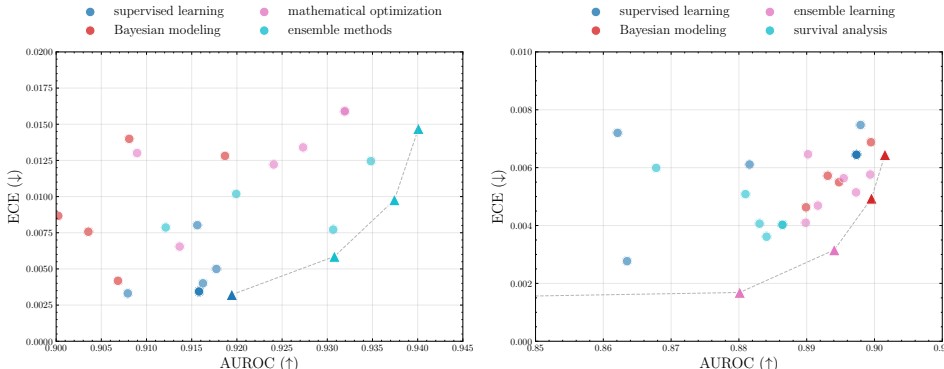

Figure 4: **NHANES/SEER Pareto fronts.** Pareto-efficient models discovered across AUROC–ECE trade-offs.

## B  EXTENDED EXPERIMENTS AND EMPIRICAL ANALYSES

### B.1  ADDITIONAL DOMAINS: LARGE-SCALE MEDICAL RISK PREDICTION

We evaluate MATHMO on two further domains (NHANES and SEER) to demonstrate generality beyond the tasks in Section 5:

- **NHANES** (Akinbami et al., 2022): A national health survey containing $86,000$ records of demographic, behavioral, clinical, and environmental covariates, where the goal is to predict risk of myocardial infarction.
- **SEER** (Ries et al., 1975): A population-level cancer registry containing $100,000$ patients with incidence, demographic, and survival information, where the goal is to predict risk for breast cancer.

These experiments are designed to evaluate: *(1)* they involve *substantively different* problem domains than our earlier tasks, *(2)* evaluation is significantly *more expensive* due to training large-scale models, and *(3)* they feature *different objective trade-offs*: here, discriminative performance (AUROC ↑) and probabilistic calibration (ECE ↓), which are central in clinical modeling. We also use a gpt5o LLM backbone to highlight that MATHMO is LLM-agnostic.

**Analysis.** Figure 4 illustrates that MATHMO consistently discovers diverse Pareto-efficient models spanning supervised learners, Bayesian models, survival analysis, and ensemble methods. These experiments show that MATHMO scales to high-cost real-world modeling settings and is able to autonomously identify high-quality trade-offs across heterogenous modeling frameworks, reinforcing generality beyond the initial set of domains.

### B.2  COMPARISON TO MATURE SOLVERS AND AUTOML BASELINES

In this subsection, we compare MATHMO discovered models with mature solvers (on TSP/JSSP) and against AutoML baselines (on Ecology/Epidemiology).

**Mature solvers.** For TSP and JSSP, we benchmark against the highly optimized Concorde solver (branch-and-cut) and OR-Tools CP-SAT. These solvers are *framework-specific* and require strong manual modeling choices (variables, constraints, heuristics, tuning). In contrast, MATHMO automatically generates models across heterogeneous frameworks, producing a full Pareto frontier rather than a single optimized point. Including traditional solvers provides the strongest possible reference point, ensuring that the solutions generated by MATHMO are *sensible and well-grounded* relative to gold-standard human-engineered baselines.

**Analysis.** Tables 3 and 4 show that MATHMO attains competitive performance while simultaneously exploring solutions across optimization, heuristics, and metaheuristics, which populate distinct regions of the Pareto frontier. These comparisons validate that MATHMO produces reasonable solutions near

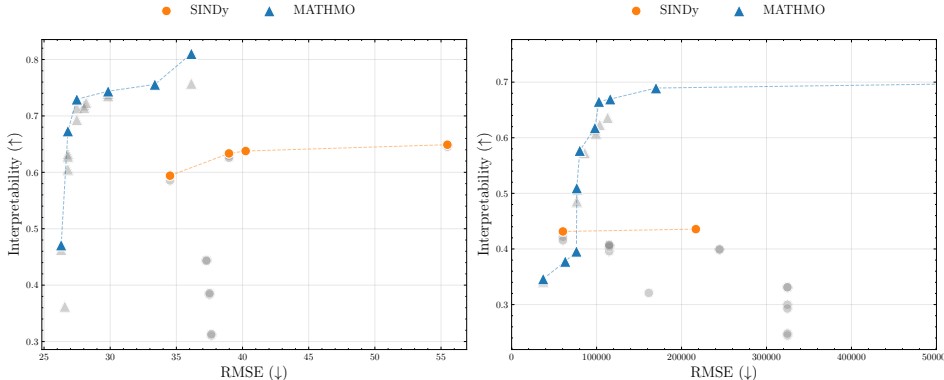

Figure 5: **SINDy vs. MATHMO.** Pareto fronts on Ecology (left) and Epidemiology (right): MATHMO dominates SINDy across accuracy and interpretability.

expert-tuned solvers, but with the added benefit of automatically discovering multiple qualitatively different trade-offs, which solvers cannot provide.

Table 3: **TSP baseline comparison.** Concorde vs. MATHMO and other baselines.

| Concorde | MATHMO | MEoH | FunSearch |
|---|---|---|---|
| (4.52, 0.0408) | (4.59, 0.6769) | (5.18, 0.1254) | (4.77, 0.1316) |
| (3.57, 0.0140) | (3.57, 0.1838) | (3.70, 0.1307) | (3.90, 0.1289) |
| (4.60, 0.0243) | (4.66, 0.6476) | (4.89, 0.1286) | (4.89, 0.1265) |
| (5.11, 0.0305) | (5.19, 0.1566) | (5.66, 0.1307) | (5.40, 0.1418) |
| (4.24, 0.0405) | (4.24, 0.1805) | (5.83, 0.1268) | (4.48, 0.1314) |

Table 4: **JSSP baseline comparison.** OR-Tools vs. MATHMO and other baselines.

| OR-Tools | MATHMO | MEoH | FunSearch |
|---|---|---|---|
| (1039.00, 0.0599) | (1039.00, 0.0230) | (1096.00, 0.0602) | (1039.00, 0.0536) |
| (1218.00, 0.2954) | (1421.00, 0.0246) | (1503.00, 0.0640) | (1372.00, 0.1610) |
| (1235.00, 13.4015) | (1235.00, 15.9644) | (1514.00, 0.0621) | (1644.00, 0.1747) |
| (1721.00, 0.6016) | (1721.00, 1.1240) | (2175.00, 0.0615) | (1925.00, 0.1697) |
| (1888.00, 0.2975) | (2127.00, 0.0233) | (2183.00, 0.0620) | (11487.00, 0.1949) |

**Comparison to SINDy.** For the time-series tasks (Ecology and epidemiology), we compare against SINDy (Brunton et al., 2016), a widely used symbolic regression technique to discover interpretable and predictive mechanistic models. As SINDy requires users to choose a basis-function library and sparsity threshold, it lends itself well to AutoML-based model selection. To keep budgets comparable, we allocate SINDy a consistent search budget of 20 configurations (4 libraries × 5 sparsity thresholds), matching the per-framework iteration budget in MATHMO.

**Analysis.** Figure 5 shows that across both Ecology and Epidemiology, SINDy is entirely dominated: its models occupy a narrow interpretability band, while MATHMO's models span a richer set of trade-offs through access to multiple modeling frameworks. **Takeaway:** MATHMO yields strictly better Pareto fronts due to its ability to explore beyond a single symbolic regression paradigm.

### B.3 COMPARISON AGAINST LLM-BASED HEURISTIC DESIGN BASELINES

We benchmark MATHMO against two recent and representative systems: **MEoH** (Yao et al., 2025) and **FunSearch** (Romera-Paredes et al., 2024). Both are state-of-the-art approaches in automated heuristic design using large language models (LLMs). Specifically, MEoH addresses multi-objective search using evolutionary operators and a dominance–dissimilarity mechanism for diversity maintenance,

while FunSearch employs genetic programming with an island-model evolutionary strategy, tailored primarily for single-objective problems. Although neither method is designed for general-purpose, cross-framework mathematical modeling, they provide strong points of reference for evaluating the relative effectiveness of `MATHMO`.

**Experimental setup.** All methods are evaluated on four benchmark problems under a fixed budget of 20 model–algorithm evaluations. For FunSearch, the multi-objective criteria are scalarized using uniform weights to ensure a fair comparison. Table 5 summarizes the results.

Table 5: Performance comparison against baselines. Higher values are better.

| Method | TSP | JSSP | Ecology | Epidemiology |
|---|---|---|---|---|
| MEoH | 0.9480 | 0.8587 | 0.8360 | 0.9054 |
| FunSearch | 0.9772 | 0.8130 | 0.6240 | 0.6628 |
| MATHMO | **0.9877** | **0.9655** | **0.9576** | **0.9793** |

`MATHMO` consistently outperforms both baselines across all domains. We attribute these gains to two main factors:

- **Cross-framework modeling.** `MATHMO` explicitly searches over diverse modeling frameworks (e.g., dynamical systems, symbolic regression, constraint programming), whereas MEoH and FunSearch tend to restrict exploration to one or two frameworks. For example, in the JSSP domain, `MATHMO` leveraged constraint programming frameworks, yielding superior solutions not discovered by either baseline.

- **Surrogate-guided search.** `MATHMO` employs LLM-based surrogate models to guide candidate selection, improving sample efficiency and focusing evaluations on high-potential areas. In contrast, MEoH and FunSearch rely exclusively on direct evaluation of evolved models.

### B.4 ROBUSTNESS AND SENSITIVITY ANALYSES

We also conducted a post-hoc sensitivity analysis by measuring the average drop in hypervolume (HV) when each framework is removed from the search space. Results are summarized below:

- **TSP:** $7.38\%$ (mathematical optimization), $7.20\%$ (metaheuristics), $1.21\%$ (heuristics)

- **JSSP:** $2.16\%$ (constraint programming), $1.29\%$ (metaheuristics)

- **Ecology:** $16.71\%$ (symbolic regression), $9.62\%$ (time-series forecasting), $0.35\%$ (rule-based)

- **Epidemiology:** $15.00\%$ (rule-based)

These results indicate that certain frameworks contribute uniquely to the Pareto frontier, while others are progressively deprioritized by the Pareto-UCB mechanism.

To further assess robustness, we conducted experiments on the vehicle routing problem (VRP) with multiple random initializations.

Table 6: Inter-run consistency on VRP (5 runs). HV denotes hypervolume.

| Run | 0 | 1 | 2 | 3 | 4 |
|---|---|---|---|---|---|
| HV | 0.989 | 0.974 | 0.980 | 0.989 | 0.987 |

Average HV: $0.984 \pm 0.005$.

The results show that hypervolume remains stable across runs ($0.984 \pm 0.005$), and framework contributions to the Pareto frontier are consistent across different initializations.

Table 7: Framework contributions to the Pareto frontier (percentage HV contribution).

| Framework | Run 0 | Run 1 | Run 2 | Run 3 | Run 4 |
|---|---|---|---|---|---|
| Heuristics | 9.02% | 2.00% | 9.54% | 8.78% | 10.21% |
| Mathematical optimization | 7.18% | 1.82% | 5.44% | 6.87% | 7.34% |
| Metaheuristics | 0.33% | — | 0.14% | 0.86% | 0.23% |

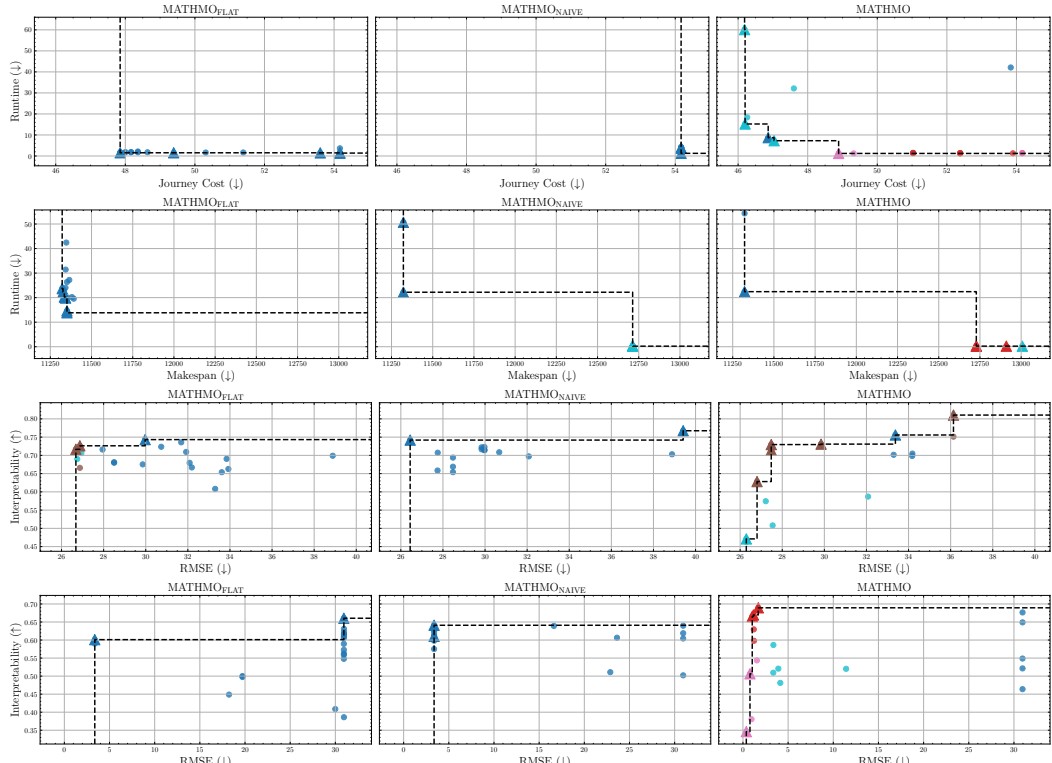

Figure 6: **Pareto front comparisons.** Visualizations of the discovered Pareto fronts for MATHMO, MATHMO$_{FLAT}$, and MATHMO$_{NAIVE}$. ▲ denotes non-dominated models; −− traces the estimated Pareto front; colors indicate distinct modeling frameworks. From top to bottom: **TSP**, **JSS**, **Ecology**, **Epidemiology**.

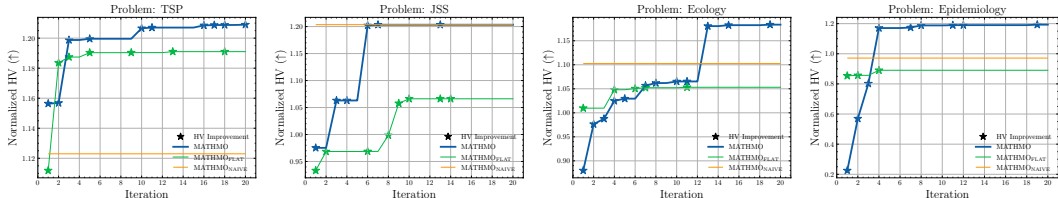

Figure 7: **Hypervolume during search.** Temporal progression of normalized hypervolume throughout the search process for MATHMO and control variants: MATHMO$_{FLAT}$ and MATHMO$_{NAIVE}$. From left to right: **TSP**, **JSS**, **Ecology**, **Epidemiology**.

## B.5 COMPARISON OF DISCOVERED PARETO FRONTS

We begin by analyzing the quality of the discovered Pareto frontiers, as well as the rate of hypervolume improvement. The Pareto front reflects the diversity and optimality of the trade-offs discovered during search, while hypervolume progression captures the efficiency with which the method explores the multi-objective space.

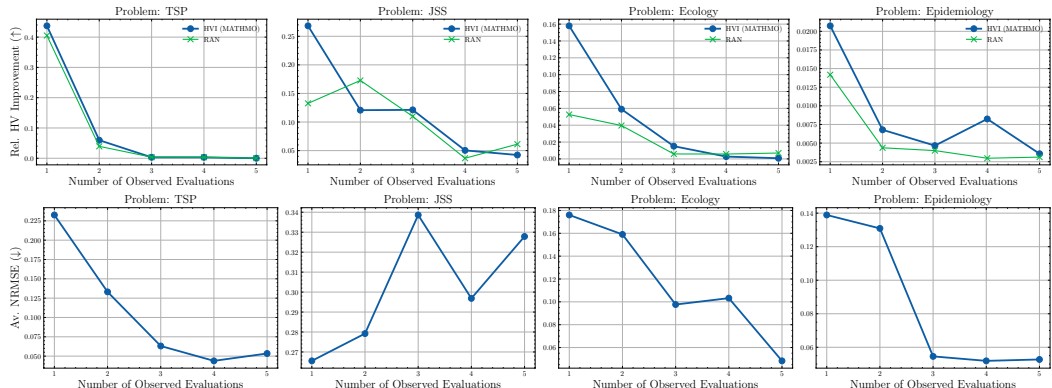

Figure 8: **Upper-level framework selection.** Comparison of hypervolume progression between two upper-level selection strategies: MATHMO (adaptive) and MATHMO_RAN (random). From left to right: **TSP**, **JSS**, **Ecology**, **Epidemiology**.

Figure 9: **Lower-level exploration efficiency. (Top)** Relative hypervolume improvement as a function of the number of model evaluations for MATHMO and RAN. **(Bottom)** NRMSE of surrogate predictions over time. From left to right: **TSP**, **JSS**, **Ecology**, **Epidemiology**

We compare MATHMO against two ablations: MATHMO_FLAT, which removes the bi-level structure and performs sequential sampling of model-algorithm pairs without adaptive framework selection; and MATHMO_NAIVE, which further removes the sequential aspect entirely, instead sampling a set of model-algorithm pairs in parallel. For all three settings, we allow 20 iterations, with MATHMO_NAIVE generating 20 parallel samples.

**Discovered pareto fronts.** In Figure 6, we compare the Pareto frontiers discovered by each method. MATHMO consistently identifies the most complete and diverse set of Pareto-efficient solutions. For instance, on **TSP**, it uncovers models that offer a range of trade-offs between journey cost and runtime. In contrast, MATHMO_FLAT explores fewer frameworks and, while it identifies some competitive models in terms of journey cost, it fails to find low-runtime alternatives with modest sacrifices in optimality. More generally, we observe that purely sequential approaches like MATHMO_FLAT tend to focus locally on just 1–2 frameworks, limiting their coverage of the multi-objective space and resulting in fewer diverse trade-off solutions. This trend is also evident in **Ecology** and **Epidemiology**, where MATHMO discovers broader and more densely populated Pareto fronts that dominate across both objectives. An exception arises in **JSS**, where MATHMO_NAIVE discovers a Pareto front comparable to that of MATHMO. This is likely due to favorable random coverage and the nature of the problem landscape, where multiple high-performing models can be sampled without requiring adaptive search.

**Search efficiency.** Figure 7 shows the progression of normalized hypervolume across iterations. On **TSP**, **Ecology**, and **Epidemiology**, MATHMO achieves faster gains in hypervolume compared to the ablations, often requiring significantly fewer iterations to identify Pareto-efficient solutions. The case of **JSS** is again an outlier: although MATHMO_NAIVE eventually achieves comparable hypervolume, MATHMO reaches the same level in only 6 iterations, highlighting its superior sample efficiency.

## B.6 INSIGHTS: UPPER-LEVEL FRAMEWORK SELECTION

We next examine the dynamics of upper-level framework selection, which in MATHMO is guided by the Pareto-UCB utility function. To isolate its impact, we compare against a control ablation,

MATHMO$_{\text{RAN}}$, which is identical to MATHMO except that it replaces utility-guided selection with uniform random sampling over frameworks.

**Observations.** Figure 8 shows the progression of normalized hypervolume across iterations. On **TSP**, **Ecology**, and **Epidemiology**, we observe similar rates of improvement between MATHMO and MATHMO$_{\text{RAN}}$ during the initial $t \leq 6$ iterations, reflecting the exploratory phase of the search. However, after this point, MATHMO exhibits a notably sustained increase in hypervolume, suggesting that it effectively focuses its search budget on frameworks with greater potential for Pareto improvement. This behavior illustrates the utility of guided upper-level selection: Pareto-UCB enables adaptive resource allocation toward promising regions of the search space.

In contrast, on **JSS**, both methods converge to comparable hypervolume levels at similar rates. This echoes the earlier observation in Pareto front comparisons, where MATHMO$_{\text{NAIVE}}$ also performed well. These results suggest that the effectiveness of upper-level selection may be problem-dependent. For instance, in settings like **JSS** where the landscape is relatively flat or where good models are distributed across many frameworks, random selection may suffice to find competitive solutions.

### B.7 INSIGHTS: LOWER-LEVEL EXPLORATION EFFICIENCY

We now turn to the efficiency of local exploration, specifically how well the surrogate model guides the selection of candidate model-algorithm pairs. To isolate this effect, we compare MATHMO against a control variant, RAN, which is identical in every respect except that it selects a candidate at random rather than using a surrogate to estimate and optimize objective performance.

To ensure a fair comparison, both methods are evaluated on the same historical set of models and the same set of candidate proposals at each step. This setup, averaged over 5 random seeds, controls for variation in the history and candidate pool, ensuring that any observed differences can be attributed solely to the decision-making strategy, i.e., surrogate-guided versus random selection.

**Local search efficiency.** Figure 9 **(Top)** shows relative hypervolume improvement as a function of the number of evaluated models. For each selected point, we compute the gain in hypervolume relative to the historical set alone; a larger value indicates that the newly acquired model improved the current Pareto front. Across all four benchmarks, **TSP**, **JSS**, **Ecology**, and **Epidemiology**, we observe that MATHMO consistently achieves higher relative hypervolume gains. This indicates that surrogate-guided selection is more effective at identifying models that advance the Pareto frontier, leading to more sample-efficient exploration compared to random selection.

**Surrogate model performance.** To better understand the surrogate's behavior, we also measure its predictive accuracy in terms of normalized RMSE (NRMSE) on the candidate models, averaged across the two objectives. In **TSP**, **Ecology**, and **Epidemiology**, we find that NRMSE decreases with more historical data, suggesting that the LLM-based surrogate improves with more observations, consistent with the observation in Liu et al. (2024) that LLM-based surrogate estimation generalizes better with more context. In contrast, on **JSS**, surrogate accuracy worsens over time, with increasing NRMSE across iterations. Manual inspection revealed a possible explanation: that the dominant source of error lies in estimating runtime for metaheuristic algorithms—critical for identifying Pareto-improving trade-offs in this problem domain. These runtime behaviors are difficult to predict based solely on surface-level descriptions, leading to poor surrogate performance. This likely explains why both MATHMO and RAN achieve similar relative hypervolume improvements on **JSS**: as the surrogate becomes less informative, its selection decisions approach random choice.

### B.8 ADDITIONAL EVALUATIONS OF MOSE

**Sensitivity analysis.** MOSE is used during search as a surrogate for subjective human preferences (e.g., model interpretability). Surrogate stability is critical: if small changes to the reference set $\mathcal{M}_{\text{ref}}$ produced inconsistent preferences, MATHMO's search could be noisy or unreliable. Therefore, evaluating the sensitivity of MOSE directly addresses whether its interpretability judgments are robust.

**Analysis.** We compute MOSE scores for 20 models across four independently sampled reference sets. The distributions in Figure 10 and correlations in Tables 8 and 9 show consistently high agreement ($r > 0.93$). MOSE produces stable and consistent judgments across reference sets, indicating that it is a reliable component for guiding subjective-objective trade-offs during search.

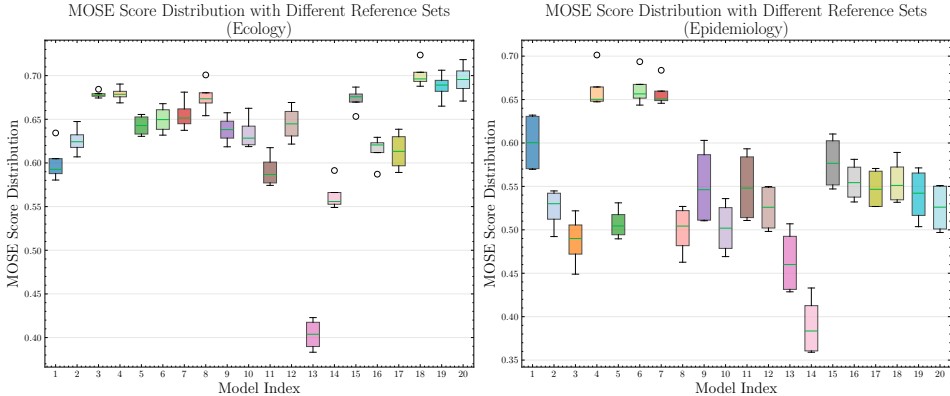

Figure 10: **MOSE sensitivity.** Interpretability score distributions across four reference sets ($\mathcal{M}_{\text{ref}}$).

**Human study.** As MOSE is used to approximate human qualitative preferences, it is essential to verify that its judgments align with human experts. The goal is not to perform a full-scale user study, but to provide evidence that MOSE is directionally consistent with expert reasoning. We collect expert pairwise interpretability judgments on 25 randomly sampled model pairs. For each pair, we record the MOSE scores and the expert preference of interpretability.

**Analysis.** Agreement with MOSE is 79.2% for Ecology (Table 10) and 76.0% for Epidemiology (Table 11). The high agreement rates indicate that MOSE captures *meaningful subjective preferences* consistent with expert intuition, supporting its use as a scalable interpretability surrogate.

Table 8: Epidemiology - Correlations Between Reference Sets ($\mathcal{M}_{\text{ref}}$)

| Reference Set Pair | Pearson $r \uparrow$ ($p$-value) |
|---|---|
| (3, 2) | 0.9695 (1.92e-12) |
| (3, 1) | 0.9801 (4.30e-14) |
| (3, 0) | 0.9673 (3.61e-12) |
| (2, 1) | 0.9892 (1.85e-16) |
| (2, 0) | 0.9507 (1.36e-10) |
| (1, 0) | 0.9561 (4.87e-11) |

Table 9: Ecology - Correlations Between Reference Sets ($\mathcal{M}_{\text{ref}}$)

| Reference Set Pair | Pearson $r \uparrow$ ($p$-value) |
|---|---|
| (3, 2) | 0.9823 (1.53e-14) |
| (3, 1) | 0.9525 (9.87e-11) |
| (3, 0) | 0.9923 (8.84e-18) |
| (2, 1) | 0.9351 (1.54e-09) |
| (2, 0) | 0.9809 (3.06e-14) |
| (1, 0) | 0.9567 (4.36e-11) |

Table 10: **Human interpretability judgments (Ecology).** Pairwise expert preferences compared against MOSE predictions for 25 model pairs. Agreement indicates whether MOSE selects the same model as the human expert.

| MOSE(A) | MOSE(B) | Expert | Agree |
|---------|---------|--------|-------|
| 0.403 | 0.673 | B | ✓ |
| 0.626 | 0.645 | A | ✗ |
| 0.600 | 0.675 | B | ✓ |
| 0.650 | 0.643 | B | ✗ |
| 0.403 | 0.679 | B | ✓ |
| 0.679 | 0.675 | A | ✓ |
| 0.626 | 0.687 | B | ✓ |
| 0.626 | 0.635 | A | ✗ |
| 0.563 | 0.687 | B | ✓ |
| 0.673 | 0.635 | A | ✓ |
| 0.626 | 0.650 | B | ✓ |
| 0.591 | 0.701 | B | ✓ |
| 0.655 | 0.614 | A | ✓ |
| 0.600 | 0.635 | B | ✓ |
| 0.600 | 0.638 | B | ✓ |
| 0.626 | 0.655 | B | ✓ |
| 0.679 | 0.650 | B | ✗ |
| 0.679 | 0.635 | A | ✓ |
| 0.673 | 0.638 | A | ✓ |
| 0.645 | 0.403 | A | ✓ |
| 0.687 | 0.635 | A | ✓ |
| 0.614 | 0.678 | B | ✓ |
| 0.403 | 0.701 | A | ✗ |
| 0.614 | 0.614 | A | ✓ |
| **Agreement Rate** | | | **19/24 (79.2%)** |

Table 11: **Human interpretability judgments (Epidemiology).** Expert pairwise evaluations compared against MOSE decisions for 25 model pairs. Agreement indicates whether MOSE matches expert preference.

| MOSE(A) | MOSE(B) | Expert | Agree |
|---------|---------|--------|-------|
| 0.464 | 0.578 | B | ✓ |
| 0.524 | 0.525 | B | ✓ |
| 0.601 | 0.500 | A | ✓ |
| 0.663 | 0.507 | A | ✓ |
| 0.464 | 0.662 | B | ✓ |
| 0.662 | 0.500 | A | ✓ |
| 0.524 | 0.540 | A | ✗ |
| 0.524 | 0.502 | B | ✗ |
| 0.390 | 0.540 | B | ✓ |
| 0.578 | 0.502 | B | ✗ |
| 0.524 | 0.663 | B | ✓ |
| 0.550 | 0.556 | B | ✓ |
| 0.658 | 0.548 | A | ✓ |
| 0.601 | 0.502 | A | ✓ |
| 0.601 | 0.551 | A | ✓ |
| 0.524 | 0.658 | B | ✓ |
| 0.662 | 0.663 | A | ✗ |
| 0.662 | 0.502 | B | ✗ |
| 0.578 | 0.551 | A | ✓ |
| 0.525 | 0.464 | A | ✓ |
| 0.556 | 0.540 | A | ✓ |
| 0.540 | 0.502 | B | ✗ |
| 0.548 | 0.488 | A | ✓ |
| 0.464 | 0.556 | B | ✓ |
| 0.555 | 0.548 | A | ✓ |
| **Agreement Rate** | | | **19/25 (76.0%)** |

## C  TECHNICAL DETAILS

In this section of the Appendix, we provide additional details on the implementation of MATHMO.

### C.1  ALGORITHM/PSEUDOCODE

---

**Algorithm 1** Bi-level Adaptive Search Loop in MATHMO

---

1: **Input:** Problem description $p$, dataset $\mathcal{D}_p$, number of iterations $T$, number of frameworks $F$, number of candidates $L$
2: Sample initial frameworks: $f_i \sim p_\theta(\cdot \mid p),\ \forall\, i \in [F]$
3: Initialize histories: $S_i^{(0)} = \emptyset,\ \forall\, i \in [F]$
4: **for** $t = 1$ to $T$ **do**
5:     **Upper-level:** Compute utility: $\alpha(f_i) = \text{Pareto-UCB}(f_i;\ S_i^{(t-1)}),\ \forall\, i \in [F]$
6:     Select framework: $f_* = \arg\max_{f_i \in \mathcal{F}} \alpha(f_i)$
7:     **Lower-level:** Sample candidate pairs: $(\tilde{m}_j, \tilde{a}_j) \sim p_\phi(\cdot, \cdot \mid p, f_*, S_*^{(t-1)}), \forall\, j \in [L]$
8:     Estimate objectives: $\hat{r}_j = p_{\text{SM}}(\tilde{m}_j, \tilde{a}_j \mid p, f_*, S_*^{(t-1)})\ \forall\, j \in [L]$
9:     Select candidate pair: $(m^{(t)}, a^{(t)}) = \arg\max_{(m,a) \in \tilde{\mathcal{C}}} \text{HV}(\hat{r}_j; r_{\text{ref}})$
10:     Solve and evaluate to obtain $r^{(t)}$
11:     Update history: $S_*^{(t)} \leftarrow S_*^{(t-1)} \cup \{(m^{(t)}, a^{(t)}, r^{(t)})\}$
12: **end for**
13: **Output:** Pareto set: $\mathcal{P} = \text{Pareto}\left(\bigcup_{i=1}^{F} S_i^{(T)}\right)$

---

### C.2  LLM SEARCH OPERATORS

To recap, MATHMO leverages LLMs as search operators, specifically for three distinct roles: sampling realizations, surrogate evaluations of candidate models, and as a model of subjective evaluation (MOSE).

1. **Generative sampler [frameworks].** Conditioned on the problem description $p$, LLMs are prompted to sample suitable modeling frameworks, which we denote as $f \sim p_\theta(\cdot \mid p)$. Specifically, the LLM is instructed to return proposed frameworks in a JSON structure containing two fields: "modeling_framework" (string) and "framework_description". The prompt skeleton and output format are described in Figures 11 and 12 respectively. Note that the descriptions enclosed in {} represent placeholder values that are populated dynamically at runtime.

2. **Generative sampler [model and algorithm].** Conditioned on the problem description $p$, a selected framework $f$, and a set of previously evaluated models within that framework $\mathcal{S}^f$, the LLM generates a new model-algorithm pair, $(m, a) \sim p_\phi(\cdot, \cdot \mid p, f, \mathcal{S}^f)$. The output is returned in a JSON format with the fields: "model" (a Python code string), "dependencies" (a list of package names), and "explanation" (a rationale for the design). Prompt details and output structure are described in Figures 13 and 14.

3. **Surrogate evaluations [candidate model].** For each candidate model-algorithm pair $(\tilde{m}, \tilde{a})$, LLMs are employed as surrogates to estimate multi-objective performance metrics, offering a low-cost approximation. This approach is motivated by the unstructured nature of the input space, which differs from traditional numerical or mixed-integer domains in Bayesian Optimization (Snoek et al., 2012), and is supported by recent successes of LLM-based surrogates in language-driven domains (Liu et al., 2024; Requeima et al., 2024). Each objective $\hat{r}_j \in \mathbb{R}$ for $j \in [k]$ is estimated independently using the LLM, based on input $(\tilde{m}, \tilde{a}, p, f, \mathcal{S}^f)$, where $\hat{r}_j = p_{\text{SM}}(\tilde{m}, \tilde{a} \mid p, f, \mathcal{S}^f)$. Multiple predictions are sampled in parallel to construct an empirical predictive distribution over the objectives. The prompt to achieve this is described in Figure 15.

4. **MOSE Surrogate Model of Subjective Evaluations.** LLMs also serve as a surrogate for subjective human judgment, enabling a generalized, cross-framework assessment mechanism for qualitative criteria. Prior work has demonstrated that LLMs can effectively model human preferences in alignment, safety, and prose diversity tasks (Bai et al., 2022; Bradley et al., 2024). In our framework, MOSE uses an LLM to predict whether a proposed model $m_t$ is subjectively preferred

over baseline models $m_i \in \mathcal{M}_{\text{ref}}$, given a problem description $p$. This is formalized as $p_{\text{MOSE}}(m_t \succ m_i \mid p)$. A prediction of '1' indicates that $m_t$ is subjectively superior. The associated token probabilities are extracted. This process is repeated for each $m_i \in \mathcal{M}_{\text{ref}}$, where the preference scores against each reference model are averaged to compute the final MOSE score. The prompt structure depicted in Figure 16.

```
  You are an expert modeling assistant. Your task is to help the user
      create a formal model to solve their problem.

  **Task:** You will receive a description of the problem and the
      desired objective(s) of the model. Your job is to propose a
      modeling framework that can be used to solve the problem. You
      should also provide a detailed explanation of your proposed
      framework, including any assumptions or constraints that you are
      making.

  **Problem description:**
  {GENERAL_PROBLEM_DESCRIPTION}

  **Problem instance descriptions:**
  {INSTANCE_DESCRIPTION}

  **Output format requirement:**
  - You must output your response as a single, valid JSON object.
  - No other text should precede or follow the JSON. The JSON object
      must strictly follow this structure:
  {OUTPUT_FORMAT_REQUIREMENT}
```

Figure 11: Prompt structure for **framework proposal**.

```
{
  "modeling_framework" (string): "concise terminology to generally
      describe the modeling framework (e.g., mathematical
      optimization, dynamical systems)",
  "framework_description" (string): "high-level description of the
      proposed modeling framework"
}
```

Figure 12: Output format for **proposed frameworks**.

## C.3 ADAPTIVE SEARCH

Having detailed the implementation of LLM search operators in MATHMO, we now cover various implementation details of the adaptive search process. Subsequently, we tabulate the key hyperparameters and describe the computational resources.

**Upper-level: framework selection.** At the beginning of the search process ($t = 0$), an initial set of $w$ candidate frameworks is proposed independently using the LLM-based sampler $f \sim p_\theta(\cdot \mid p)$. Then in each iteration:

1. **Compute statistics.** For each framework, we compute summary statistics using the framework-specific historical performance vectors $\{r_t \mid (m_t, a_t, r_t) \in \mathcal{S}^f\}$, specifically the empirical mean $\mu_f \in \mathbb{R}^k$ and variance $\sigma_f^2 \in \mathbb{R}^k$.
2. **Compute UCB.** As our $\alpha(\cdot, \cdot)$ is implemented using the Pareto-UCB policy, we compute the UCB vector $\text{UCB}_f \in \mathbb{R}^k$ for each framework using Equation (2).
3. **Identify Pareto set.** Using the set of UCB vectors, the set of Pareto optimal (non-dominated) frameworks is identified.

```
    You are an expert modeling assistant. Your task is to help the user
        create a formal model to solve their problem.

    **Task:** You will receive a description of the problem and the
        desired objective(s) of the model. Your job is to return the
        model that can generate the required output/solution in the
        output format specified. The model you generate should belong to
        the modeling framework specified.

    **Problem description:**
    {GENERAL_PROBLEM_DESCRIPTION}

    **Problem instance descriptions:**
    {INSTANCE_DESCRIPTION}

    **Chosen modeling framework:**
    {MODELING_FRAMEWORK}

    **Output format requirements:**
    {OUTPUT_FORMAT_REQUIREMENTS}
```

Figure 13: Prompt structure for **model/algorithm proposal**.

```
{
  "model" (Python code): "complete Python code of the model and
      algorithm generated to represent and solve the provided problem
      .",
  "dependencies" (list): "list of external Python package dependency"
      ,
  "model_explanation" (str): "detailed description of the generated
      model and algorithm"
}
```

Figure 14: Output format for **proposed model/algorithm**.

4. **Selection.** If there exists more than one framework in the Pareto-UCB set, one framework is randomly selected to be explored next.

**Lower-level: local exploration.** In each iteration of lower-level exploration, the following steps occur:

1. **Proposal.** A set of candidate model-algorithm pairs are sampled, denoted as $\tilde{\mathcal{S}}^f = \{\tilde{m}^{(i)}, \tilde{a}^{(i)} \,|\, i \in [l]\}$.
2. **Surrogate estimation.** For each candidate pair, we obtain an estimated objective vector $\hat{r}^{(i)} = p_{\text{SM}}(\cdot \,|\, \tilde{m}^{(i)}, \tilde{a}^{(i)}; p, f, \mathcal{S}^f)$.
3. **Selection.** The pair $(m_t, a_t)$ that yields the largest estimated hypervolume improvement $(m, a) = \arg\max_{(\tilde{m}, \tilde{a}) \in \tilde{\mathcal{S}}^f} \text{HV}(\tilde{m}, \tilde{a}; r_{\text{ref}})$ is selected to undergo evaluation.
4. **Evaluation.** We execute the model and algorithm as a subprocess. If any errors occurred during execution, the error trace is extracted and passed to the LLM to fix any mistakes and regenerate the model. If the model still does not execute after 3 `MAX_RETRIES`, the process returns to the proposal step. Note that there is also a 300 second `TIMEOUT` imposed on model execution. If the model timeout, it is forcefully terminated.
5. **Observe objectives.** The objective vectors $r \in \mathbb{R}^k$ are obtained by evaluating the model and generated outputs, and the triplet are added to the set of models $\mathcal{S}^f \leftarrow \mathcal{S}^f \cup (m, a, r)$.

### C.4 MISCELLANEOUS IMPLEMENTATION DETAILS

**Key hyperparameters.** We detail the hyperparameters for implementing `MATHMO` in Table 12.

```
You are an expert evaluation assistant. Your task is to help the user
    evaluate models that were generated for a particular problem.

**Task:** Your role is to help evaluate mathematical models designed
    for a specific problem. You will be given:
- A description of the problem, the objective(s) the models are
    intended to achieve,
- A history of previously generated models along with their
    performance metrics,
- A candidate model for evaluation.
Based on this information, predict the likely performance of the new
    candidate model.

**Problem description:**
{GENERAL_PROBLEM_DESCRIPTION}

**Problem instance descriptions:**
{INSTANCE_DESCRIPTION}

**History of proposed models:**
{MODEL_HISTORY}

**Output format requirements:**
{OUTPUT_FORMAT_REQUIREMENTS}
```

Figure 15: Prompt structure for **surrogate evaluations**.

```
You are an expert at evaluating the subjective quality of models.
    Your task is to assess the subjective quality of a model based on
    its description against the baseline reference provided.

**Task:** You will receive a target model description (in code) and a
    baseline model description (also in code). Your job is to assess
    the {CRITERION} of the target model compared to the baseline
    reference. Provide a detailed assessment using the specified
    output format.

**Subjective criterion assessed:**
{CRITERION_DESCRIPTION}

**Important instructions:**
- Your assessment/explanation should be grounded in semantic meaning
    of the target model and the baseline reference.
- The assessment should be based on your best intuition and semantic
    understanding of the models.
- Then score the target model, returning 1 if the target model is
    more {CRITERION_VALUE} than the baseline reference model, or 0 if
    not.

**Target model:**
{TARGET_MODEL}

**Baseline reference model:**
{BASELINE_REF}
```

Figure 16: Prompt structure for **surrogate evaluations**.

**LLM.** We use `gpt-4o-2024-05-13` as the underlying LLM in all experiments.

**Computer resources.** We run all experiments on an `AMD EPYC 7V13 64-Core Processor`.

**Code and reproducibility.** The full implementation of `MATHMO`, along with the code necessary to reproduce all key results, will be released on GitHub upon acceptance of the paper.

Table 12: **Description of key hyperparameters.**

| Hyperparameter | Description | Value |
|---|---|---|
| $w$ | Number of frameworks | 4 |
| $l$ | Number of candidate model-algorithm pairs proposed in each iteration of local exploration | 3 |
| $q$ | Number of MC estimates for surrogate evaluations | 3 |
| $(c, d)$ | Pareto-UCB exploration bonus hyperparameters | $(1, 1)$ (default) |
| $|\mathcal{M}_{\text{ref}}|$ | Number of baseline models used in `MOSE` evaluations | 3 |
| $T$ | Number of search iterations | 20 |
| $\tau$ | LLM hyperparameter (sampling temperature) | 0.7 (default) |
| $p$ | LLM hyperparameter (top-p sampling) | 0.9 (default) |
| `TIMEOUT` | Max runtime (in seconds) allowed for each model-algorithm pair to execute | 300 |

## D    EXPERIMENTAL DETAILS

In this section of the appendix, we will describe the datasets and metrics employed in our empirical evaluations.

### D.1    DATASETS

**Traveling Salesman (TSP).** The Traveling Salesman Problem (TSP) is a foundational combinatorial optimization problem where the objective is to find the shortest possible route that visits a set of cities exactly once and returns to the starting point. The two primary trade-offs in modeling TSP are **journey cost** (total tour length) and **runtime** (solution time). Journey cost reflects the quality of the solution and is crucial in applications like logistics and manufacturing, while runtime is vital in scenarios requiring rapid decisions, such as dynamic routing. Common modeling techniques include: (1) exact methods such as Integer Linear Programming (ILP), which guarantee globally optimal solutions but scale poorly with problem size; (2) metaheuristic approaches like Genetic Algorithms, which offer faster approximate solutions at the cost of optimality; and (3) domain-specific heuristics such as nearest-neighbor or insertion algorithms, which are simple and computationally efficient (Bellmore and Nemhauser, 1968). For our experiments, we generated 10 random Euclidean instances with 30–50 cities each. Each instance was created by uniformly sampling city coordinates in a 2D unit square, a standard method for generating synthetic TSP datasets.

**Job Shop Scheduling (JSS).** Job Shop Scheduling is a canonical operations research problem that involves assigning a sequence of jobs to a set of machines, where each job consists of a series of operations with specific processing requirements. The primary objectives are to minimize the **makespan** (i.e., the total time required to complete all jobs) and to reduce **runtime**, which becomes critical in dynamic or large-scale industrial systems. A lower makespan increases throughput, directly impacting productivity, while efficient computation ensures that schedules can be adapted in real time. (1) Constraint Programming (CP), which provides precise encodings but may struggle with scalability; (2) metaheuristic methods such as Tabu Search or Simulated Annealing, which balance exploration and exploitation to find high-quality solutions efficiently; and (3) greedy or rule-based heuristics, which offer speed and interpretability at the cost of optimality (Xiong et al., 2022). We use 10 well-known benchmark instances from Lawrance (1984), which are widely adopted in the scheduling literature. These instances span 10 to 30 jobs, 5 to 10 machines, and 50 to 300 operations. The dataset is available through the open-source `job_shop_lib` repository (https://github.com/Pabloo22/job_shop_lib).

**Ecology.** Ecological modeling seeks to understand and predict interactions among species and their environments, often involving dynamic systems such as predator-prey relationships. A key modeling trade-off in ecology lies between **predictive performance**—capturing future population dynamics accurately—and **interpretability**, which is critical for gaining ecological insights and informing conservation efforts. Common modeling approaches include: (1) differential equation systems such as Lotka-Volterra models, which provide interpretable representations of species interactions; (2) classical time-series models like ARIMA, which are effective for short-term forecasts; and (3) probabilistic graphical models, which capture structured uncertainty and latent ecological processes (van den Berg et al., 2022). We use a dataset containing Hare-Lynx populations (Stenseth et al.,

1997), which records annual observations of the Snowshoe Hare and Canadian Lynx populations over multiple decades.

**Epidemiology.** Epidemiological modeling focuses on understanding and forecasting the spread of infectious diseases, often under constraints that demand both accurate **prediction** and clear **interpretability** for public health decision-making. Predictive performance ensures that interventions can be timed effectively, while interpretability allows stakeholders to understand transmission mechanisms and policy implications. Common approaches include: (1) compartmental models (e.g., SIR, SEIR), which capture the flow of individuals through disease states using differential equations; (2) statistical models such as Poisson and negative binomial regressions, which model count data under uncertainty; and (3) time-series forecasting techniques, including autoregressive and neural models, for flexible temporal prediction (Xiang et al., 2021). For our experiments, we use COVID-19 time series data from Italy, sourced from the COVID-19 Data Repository by the Center for Systems Science and Engineering (CSSE) at Johns Hopkins University (Dong et al., 2020). The dataset contains daily counts of confirmed cases, deaths, and recoveries.

We note that in all our experiments, the dataset is provided as an input only at runtime—after the model has been generated by the LLM. The LLMs themselves do not have access to the dataset contents during model generation; they are only given high-level metadata, such as the number of features, problem size (e.g., number of operations or cities), or time series length, to inform their proposals.

## D.2 METRICS

**Hypervolume.** The hypervolume (HV) metric quantifies the volume of the objective space that is dominated by a set of solutions, relative to a fixed reference point. It serves as a standard measure in multi-objective optimization, capturing both convergence and diversity of the solution set. Formally, let $\mathcal{R} = r_1, \ldots, r_n$ be a set of $n$ $k$-dimensional objective vectors and let $r_{\text{ref}} \in \mathbb{R}^k$ be a reference point that is dominated by all vectors in $\mathcal{R}$. The hypervolume is defined as:

$$\text{HV}(\mathcal{R}; r_{\text{ref}}) = \lambda\Big( \bigcup_{r \in \mathcal{R}} [r_1, r_{\text{ref},1}] \times \cdots \times [r_k, r_{\text{ref},k}] \Big) \tag{5}$$

where $\lambda$ denotes the Lebesgue measure in $\mathbb{R}^k$. All objectives are first normalized to $[0, 1]$, and we set $r_{\text{ref}} = 1.1$ to ensure it lies outside the normalized Pareto front. We compute hypervolume using the `pymoo` library (Blank and Deb, 2020) (https://pypi.org/project/pymoo/).

**Relative Hypervolume Improvement.** To assess progress over time, we compute the relative hypervolume improvement, which quantifies the gain in hypervolume relative to the best value achieved at a previous timestep. Let $\text{HV}_t$ and $\text{HV}_{t'}$ denote the HV at iteration $t$ and $t'$, where $t' < t$.

$$\text{RHI}_t = \frac{\text{HV}_t - \text{HV}_{t'}}{\text{HV}_{t'}} \tag{6}$$

We employ this metric to compare the impact of search strategies over the course of search.

**Normalized RMSE.** Root Mean Squared Error (RMSE) is a standard regression metric that measures the average magnitude of prediction error. In our context, we use a normalized RMSE (NRMSE) to account for scale differences across objectives. Given a set of ground-truth values $\{y_i\}_{i=1}^n$ and predictions $\{\hat{y}_i\}_{i=1}^n$, RMSE is defined as:

$$\text{RMSE} = \sqrt{\frac{1}{n} \sum_{i=1}^{n} (y_i - \hat{y}_i)^2} \tag{7}$$

We normalize this by the empirical range of the true values $\sigma_y$, yielding:

$$\text{NRMSE} = \frac{\text{RMSE}}{\sigma_y} \tag{8}$$

**Permutation entropy.** Permutation entropy (PE) is a model-free measure of complexity for time series or ordered sequences, capturing the unpredictability of local ordering patterns. Given a time series $\{x_t\}_{t=1}^T$, the sequence is partitioned into overlapping windows of length $d$ (embedding

dimension), and each window is mapped to a permutation pattern based on the relative ordering of its elements. Let $\pi_i$ denote the $i$-th unique pattern and $p(\pi_i)$ its empirical frequency. The PE is then defined as:

$$H_d = -\sum_i p(\pi_i) \log p(\pi_i) \tag{9}$$

which is often normalized by $\log(d!)$ to yield a value in $[0, 1]$. We compute PE using the `antropy` library (https://pypi.org/project/antropy/). This metric provides a lens into the structural complexity of sequences produced by time-series models

