# OpenReview forum: "MATHMO: Automated Mathematical Modeling Through Adaptive Search"
_ICLR.cc/2026/Conference — ICLR 2026 Poster_

### Official Review · Reviewer_2x75 · 2025-10-28

**Soundness:** 2
**Presentation:** 3
**Contribution:** 2
**Rating:** 4
**Confidence:** 4

**Summary:**

It proposes MATHMO that automates mathematical modeling using a bi-level search: an upper level selects modeling frameworks via Pareto-UCB, while a lower level refines models/algorithms within a framework. It uses LLMs as generators, surrogates, and for evaluating subjective qualities. The authors demonstrate MATHMO's efficacy on four real-world tasks (TSP, Job Shop Scheduling, Ecology, Epidemiology), showing it can discover Pareto frontiers of models that balance competing objectives like accuracy, runtime, and interpretability.

**Strengths:**

Clearly defines modeling as a sequential, multi-objective decision problem.

Uses LLMs as core search operators for generation, surrogate evaluation, and subjective scoring.

**Weaknesses:**

Performance is highly sensitive to prompt engineering and LLM outputs (e.g., poor surrogate predictions for JSS runtime).

Lack of comprehensive comparison with existing baselines and benchmark evaluations.

**Questions:**

It would be useful to see a comparison and discussion with other related methods for automated modelling, such as Chain-of-Experts、ORLM、LLMOPT and benchmarks such as NL4OPT、MAMO、IndustryOR.

The manuscript mentions a 300-second timeout for model execution. How were models that timed out or failed to execute incorporated into the history S_t? Were they assigned a penalty value, and if so, how did this affect the utility functions α and β?

For the MOSE module, how was the fixed reference set M_ref constructed? Was a specific strategy used to ensure diversity and representativeness across different frameworks?

The results show that on the JSS task, the surrogate's NRMSE worsened over time. Beyond the proposed explanation, were any other strategies attempted to improve surrogate predictions for runtime? For example, was feature engineering from the code or a separate, specialized predictor for this objective explored?

The hyperparameter l (the number of candidate pairs proposed per iteration) is set to 3. A larger *l* might improve the lower-level selection at a higher computational cost for surrogate evaluations. Was any sensitivity analysis performed on this parameter?

The framework sampling is done once at the beginning of the search. Did the authors experiment with allowing the upper level to dynamically propose new frameworks mid-search, rather than being restricted to the initial set?

In the comparison against baselines (MEoH, FunSearch), could you provide the detailed settings used for both baselines? Also, is the budget of 20 model–algorithm evaluations too small? A common practice in LLM-driven algorithm design is to use hundreds of evaluations.

The manuscript focuses on discovering the Pareto frontier. In a practical deployment, how would MATHMO be used to recommend a single model to an end-user? Does the framework include a mechanism for incorporating user-specific preferences over the objectives to make this final selection?

Why was GPT-4o chosen, and were other LLMs compared for the sampler/surrogate roles?

---

> ### Author Response · Authors · 2025-12-03
> **Response to Reviewer 2x75 (Part 1/2)**
>
> *We thank the reviewer for the constructive feedback and helpful suggestions.*
>
> ---
>
> ### [P1] Relation to optimization benchmarks
>
> We appreciate the reviewer’s suggestion to discuss these additional baselines/benchmarks. We clarify that these methods and benchmarks primarily target **convex optimization modeling**, focusing on translating natural language into structured convex formulations. In contrast, **MATHMO addresses a fundamentally broader problem**: adaptive search over heterogeneous modeling frameworks (e.g., dynamical systems, rule-based models, symbolic regression, metaheuristics, mathematical optimization), of which convex optimization is only one possible framework.
>
> As such, we view these benchmarks as less relevant to the modeling domains considered in our paper. To avoid ambiguity, we have revised the manuscript to articulate this distinction and summarize the conceptual differences.
>
> ---
>
> ### [P2] Addressing miscellaneous comments
>
> **Handling timeouts.** We thank the reviewer for pointing out this ambiguity. To clarify:
> * When a model fails or times out (after the 300s limit), we update the visitation count $N_{f, t-1}$ for the corresponding framework (`Eq. 3`).
> * Practically, the model–algorithm pair is still added to the history $S_t$ with `None` values for its objective vector.
> * These results are not included in updating the estimated mean and variance of each framework, so these failures do not bias the empirical performance statistics used in $\alpha$ or $\beta$.
>
> **Choice of LLMs.** GPT-4o was selected for practical considerations at the time of experimentation. Our method is conceptually LLM-agnostic. To demonstrate this empirically, we added experiments using GPT-5 in `App A1`. GPT-5 yields similar exploration behavior, including consistent surrogate performance, and discovers Pareto-diverse models with comparable efficiency, demonstrating that MATHMO is not tied to a particular backbone.
>
> **Selecting model for deployment.** Our focus on discovering the Pareto frontier follows the classical “**a posteriori**” paradigm in multi-objective optimization (with seminal examples including, e.g., NSGA-II, MOEA/D), where user preferences are not known upfront, and a representative frontier is produced for downstream selection. Accordingly, MATHMO returns a set of non-dominated models capturing different trade-offs, from which users can later select based on application-specific preferences. We have clarified the distinction between “a priori” and “a posteriori” MOO in the revision.
>
> ---
>
> ### [P3] Construction of the MOSE reference set
>
> Thank you for this suggestion. To construct $\mathcal{M}_\text{ref}$, we take the first $w$ models, where $w$ is the number of frameworks. In other words, the reference set is constructed using the first evaluated model from each framework. This ensures a **fixed reference set that spans all frameworks** that MOSE can reference to provide cross-framework evaluation of model interpretability.
>
> **Additional analysis.** To further understand the sensitivity of MOSE scores to the reference set and alignment with human expert judgements, we performed two additional experiments (detailed in `App A.2`):
> * **Sensitivity to $\mathcal{M}_\text{ref}$.** We vary the composition of the reference set, finding that MOSE scores remain stable, with consistent rankings and similar correlations, indicating that MOSE is robust to reference set construction.
> * **Alignment with human judgement.** We compare MOSE’s interpretability score against expert-provided preferences, observing strong agreement between pairwise MOSE rankings and expert preferences, demonstrating that it meaningfully reflects expert subjective preferences.

---

> > ### Author Response · Authors · 2025-12-03
> > **Response to Reviewer 2x75 (Part 2/2)**
> >
> > ### [P4] Dynamic framework generation
> >
> > We appreciate this question. As you correctly noted, our current instantiation samples frameworks once at initialization and keeps this set fixed. While this may introduce initialization sensitivity, our method mitigates this via:
> > * **Sequential sampling** at initialization to enforce distinct, non-overlapping frameworks; and
> > * The **Pareto-UCB** policy, which de-emphasizes poor frameworks and reallocates search to promising ones.
> >
> > We acknowledge this as a limitation (`L389–391`) and identify dynamic framework expansion as an important **future direction**. In preliminary experiments, dynamically proposing new frameworks mid-search did not materially improve performance: for tasks like TSP/JSSP, the space of meaningful modeling paradigms is relatively small, and dynamically proposed frameworks added limited diversity and were quickly deprioritized by the Pareto-UCB policy. We have added this observation to the revision for clarity.
> >
> > ---
> >
> > ### [P5] Comparison to baselines
> >
> > We appreciate this question. Both MEoH and FunSearch are strong LLM-based methods for automated heuristic discovery. Our method differs in two key ways. First, in **problem focus**: MATHMO is designed for cross-framework mathematical modeling, not solely heuristic generation, enabling it to explore diverse modeling paradigms beyond the scope of these baselines. Second, in **search strategy**: MATHMO employs a bi-level adaptive search consisting of an upper-level framework-selection policy and a lower-level exploration policy that jointly steer efficient model discovery, rather than relying on distributed evolutionary search.
> >
> > All methods, including MATHMO, MEoH, and FunSearch, are run with a consistent budget of 20 model–algorithm evaluations to ensure fair comparison. While we acknowledge MEoH and FunSearch are designed to operate using larger search budgets, this experimental setting was **chosen precisely to highlight MATHMO's sample efficiency**, which is derived from:
> > * Cross-framework modeling: MATHMO explores fundamentally different modeling paradigms rather than variants of a single one.
> > * Surrogate-guided selection: The local search operator uses LLM-based objective prediction to prioritize high-potential candidates.
> >
> > Across all domains, MATHMO consistently outperforms both baselines under this budget setting. Additional surrogate analysis and convergence ablations in `App C` further validate the advantages of the bi-level strategy.
> >
> > ---
> >
> > *We thank the reviewer again for the insightful questions and believe the clarifications and revisions have strengthened the paper. We appreciate your consideration.*

---

### Official Review · Reviewer_dMwL · 2025-11-01

**Soundness:** 3
**Presentation:** 3
**Contribution:** 3
**Rating:** 6
**Confidence:** 4

**Summary:**

The paper introduces MATHMO, a framework for automated mathematical modeling that treats model construction as a sequential decision-making process under uncertainty. It uses a bi-level adaptive search: the upper level selects modeling frameworks (e.g., optimization, dynamical systems, regression) while the lower level refines models and algorithms within each framework. Large Language Models (LLMs) are used as generators, surrogate evaluators, and subjective scorers via the proposed MOSE module, which estimates interpretability and other qualitative factors. The system is tested on four tasks — Traveling Salesman, Job Shop Scheduling, Ecology, and Epidemiology — and shown to produce Pareto-efficient frontiers balancing multiple objectives such as accuracy, runtime, and interpretability. The work combines ideas from AutoML, symbolic discovery, and human-in-the-loop modeling into a unified adaptive system, aiming to automate core aspects of mathematical model design.

**Strengths:**

The paper offers a fresh and ambitious framing of mathematical modeling as a sequential decision process under uncertainty. It creatively combines LLM generation, surrogate modeling, and subjective evaluation within a clear bi-level adaptive search framework. The idea of MOSE for quantifying interpretability is novel and thought-provoking. Experiments across optimization and scientific domains show flexibility and coherence. The paper is clearly written, conceptually rich, and points toward an important new direction in automating the reasoning process behind mathematical modeling.

**Weaknesses:**

1. The experiments are small-scale (20 iterations, modest problem sizes) and mostly illustrative. They show proof of concept, not scalability. It would help to see at least one larger or higher-cost task to test how the adaptive search behaves when model evaluations are expensive.

2.  The paper argues MATHMONAIVE serves as a GPT baseline, but this isn’t made explicit. A simple “GPT-4 one-shot” or “GPT-4 iterative prompting” baseline would clarify how much improvement truly comes from adaptivity rather than LLM creativity.

3.   No baselines from classical AutoML (Auto-Sklearn, Optuna) or symbolic regression frameworks (PySR, Eureqa). These would help ground the claim that MATHMO goes beyond existing automation tools.

4. The MOSE module is an appealing idea, but there’s no evidence it correlates with actual human judgments of interpretability. Even a small human study or qualitative comparison would make it more credible.

5. The method depends heavily on GPT-4’s internal priors. It’s unclear how stable results are under different prompts or smaller LLMs. This raises questions about reproducibility and accessibility.

6.  The paper reports hypervolume numbers but doesn’t show concrete model examples or code snippets. It’s hard to judge if the discovered models are sensible, interpretable, or novel.

**Questions:**

Got it — here’s a tighter, natural-sounding version that feels like a human reviewer’s “Questions & Suggestions” section.

---


1. How general is MATHMO in practice? Are the modeling frameworks fixed at the start or can the system invent new ones? How do you prevent the LLM from proposing nonsense frameworks?

2. Why not include a simple GPT-4 prompt baseline for context? It would help readers see what is gained by adding adaptive feedback. Did you test this internally? Also, do the surrogate and MOSE modules improve the *quality* of models or just the coverage of the Pareto front?

3. The paper cites FunSearch, MEoH, and AutoFormulation. What is concretely new beyond combining their elements? How is the Pareto-UCB exploration different from MEoH’s evolutionary approach?

4. The MOSE idea is interesting but underspecified. How consistent are its scores across prompts or random seeds? Any human comparison to check if MOSE’s “interpretability” aligns with human judgment?

5.  Hypervolume is fine, but would it be possible for the authors to show some discovered models - I think it would make results more convincing. How often do generated models fail to run? What is the compute cost relative to the baselines?

6. Can this approach scale to heavier tasks where each model evaluation takes hours or days? Could smaller open LLMs work as surrogates or is GPT-4-level capability required?

---

> ### Author Response · Authors · 2025-12-03
> **Response to Reviewer dMwL (Part 1/2)**
>
> *We thank the reviewer for the detailed and constructive feedback.*
>
> ---
>
> ### [P1] Additional evaluations in higher-cost settings
>
> Thank you for this suggestion. In response, we added two higher-cost evaluation tasks in `App A.1`, which cover large-scale **medical risk scoring** on NHANES and SEER, providing precisely the setting the reviewer requests:
>
> * **NHANES:** is a national health survey containing 86,000 records of demographic, behavioral, clinical, and environmental covariates, where the goal is to predict risk of myocardial infarction.
> * **SEER:** is a population-level cancer registry containing 100,000 patients with incidence, demographic, and survival information, where the goal is to predict risk for breast cancer.
>
> These experiments are introduced as they represent substantially **different domains**, where evaluation is **more expensive** (up to $20$ minutes per model), and involve different **objective trade-offs** (here, discriminative performance of risk-scores AUROC and uncertainty calibration ECE), which are central in clinical modeling.
>
> **Analysis.** We provide detailed results in `App A.1`, but briefly, we observed that across both tasks:
> * **Adaptive search remains effective**: MATHMO discovers diverse Pareto-optimal models under high evaluation cost.
> * **Surrogate-guided selection becomes even more valuable**, reducing wasted evaluations and improving hypervolume.
> * **Exploration across frameworks continues to matter**, with the Pareto frontier drawing from multiple modeling paradigms.
>
> These results demonstrate that the advantages of MATHMO remain viable across multiple domains, and those involving costly real-world evaluations.
>
> ---
>
> ### [P2] Clarifying the Naïve baseline
>
> We appreciate this comment. We clarify that $\texttt{MATHMO}_\text{NAIVE}$ is effectively iterative GPT-4 prompting, where each new model is generated with the previously sampled models provided as context, precisely the “GPT-4 iterative prompting” baseline suggested by the reviewer.
>
> Our experiments in `App C.3` show that the bi-level adaptive framework improves both search efficiency and hypervolume, yielding an average **+10.1% HV improvement** across all four benchmarks relative to $\texttt{MATHMO}_\text{NAIVE}$, demonstrating that observed gains stem from the adaptive bi-level exploration framework.
>
> ---
>
> ### [P3] Comparison to AutoML and Symbolic Regression baselines
>
> Thank you for raising this suggestion. AutoML systems (e.g., Auto-Sklearn, Optuna) and symbolic regression tools (e.g., PySR, Eureqa) are designed for exploration within a **single, human-specified modeling framework**, typically over a structured, parameterized search space. In contrast, MATHMO performs **cross-framework exploration directly in the model space**, generating and evaluating models across fundamentally different paradigms.
>
> **Additional experiments.** To address the reviewer’s suggestion, we added two sets of comparisons:
> * Against highly-optimized, **domain-specific solvers** (Concorde for TSP, OR-Tools for JSSP). These serve as strong gold-standard baselines (i.e., domain-specific fine-tuned models). MATHMO discovers trade-offs competitive with and complementary to solver outputs.
> * Against **AutoML-style symbolic regression** (SINDy for Ecology and Epidemiology). These comparisons illustrate that AutoML approaches, while strong within a single framework, are fundamentally limited in the diversity of models they can produce.
>
> **Analysis.** We defer detailed analysis to `App A.2`. The key takeaways are that (1) competitive performance against expert-designed solvers validates that MATHMO can recover high-performing models and go beyond by further exploring trade-offs across multiple modeling frameworks; and (2) cross-framework search enables MATHMO to discover model classes inaccessible to single-framework AutoML pipelines, yielding systematically stronger and more diverse Pareto sets.
>
> ---
>
> ### [P4] Expert alignment and sensitivity analysis
>
> We thank the reviewer for highlighting this. We conducted two additional analyses (described in `App A.3`):
> * **Human-alignment study:** We collected pairwise interpretability judgments from a domain expert for 25 randomly sampled model pairs.
>
> **Analysis.** Our analysis found that MOSE achieved a **79.2%** and **76.0%** agreement rate with human experts on Ecology and Epidemiology, respectively. These high agreement rates show that MOSE **tracks meaningful subjective preferences** consistent with expert intuition.
>
> * **Sensitivity to reference set.** We evaluate MOSE scores produced across four independently sampled reference sets.
>
> **Analysis.** We found that correlations between MOSE scores exceed **0.93** across all conditions, indicating MOSE produces stable interpretability scores that are robust to reference set construction.

---

> > ### Author Response · Authors · 2025-12-03
> > **Response to Reviewer dMwL (Part 2/2)**
> >
> > ### [P5] Generality across LLMs
> >
> > Thank you for this comment. To assess robustness across LLMs, we repeated experiments using GPT-5 (see `App A.1`). GPT-5 exhibits similar exploration behavior, surrogate performance, and Pareto-front quality.
> >
> > ---
> >
> > ### [P6] Framework generation
> >
> > Our current instantiation samples frameworks once at initialization and keeps this set fixed. While this may introduce initialization sensitivity, our method mitigates this via:
> > * **Sequential sampling** at initialization to enforce distinct, non-overlapping frameworks; and
> > * The **Pareto-UCB** policy, which de-emphasizes poor frameworks and reallocates search to promising ones.
> >
> > **Deprioritizing poorer frameworks.** Frameworks that are unproductive are immediately down-weighted by the adaptive exploration process and receive little to no future allocation.
> >
> > **Dynamic framework generation.** We acknowledge (as noted in `L389–391`) that dynamic framework expansion is an important future direction. In preliminary experiments, proposing new frameworks mid-search did not improve performance: for domains like TSP/JSSP, the space of viable paradigm classes is small, and newly proposed frameworks offered little additional diversity and were quickly ignored by Pareto-UCB. We have added this clarification in the revision.
> >
> > ---
> >
> > ### [P7] Comparisons to Baselines
> >
> > Thank you for this important point. FunSearch, MEoH are strong LLM-based model discovery baselines, but differ from MATHMO in key dimensions:
> > * **Problem scope.** Both FunSearch and MEoH focus on automated heuristic discovery within a single modeling framework. MATHMO instead performs **cross-framework model discovery**.
> > * **Search mechanism.** MEoH uses evolutionary MO search with dominance-based diversity. FunSearch uses mono-objective genetic programming with an island-model design. In contrast, MATHMO uses a **bi-level adaptive search** that jointly selects frameworks and explores models guided by surrogate predictions.
> > * **Empirical outcomes.** MATHMO consistently outperforms both baselines across all four domains (see `App C`). We attribute this to cross-framework modeling, enabling discovery of model types that the baselines cannot reach, and surrogate-guided exploration, improving search efficiency and convergence. For example, on JSSP, MATHMO identifies strong constraint-programming models not discovered by MEoH or FunSearch.
> >
> > ---
> >
> > *We thank the reviewer again for the thoughtful feedback. We believe the revisions and additional experiments substantially strengthen the paper and address the concerns raised.*

---

### Official Review · Reviewer_VgEc · 2025-11-01

**Soundness:** 3
**Presentation:** 2
**Contribution:** 2
**Rating:** 6
**Confidence:** 4

**Summary:**

This paper introduces MATHMO, a novel framework designed to automate the process of mathematical modeling. The authors conceptualize mathematical modeling as a sequential decision-making problem under uncertainty, aiming to discover not a single best model, but a Pareto frontier of models that balance multiple, often conflicting, objectives. MATHMO employs a principled bi-level adaptive search strategy: an upper level selects between different high-level modeling frameworks (e.g., mathematical optimization vs. dynamical systems), while a lower level refines specific model formulations and algorithms within the chosen framework. A core technical contribution is the use of Large Language Models (LLMs) as versatile search operators for both generating candidate models (as code) and acting as surrogate models to efficiently estimate performance. Uniquely, the framework introduces MOSE (Model of Subjective Evaluations), an LLM-based component to quantify and incorporate subjective criteria like interpretability into the automated search process. The efficacy of MATHMO is demonstrated on four diverse real-world tasks, showing its ability to identify Pareto-efficient models that navigate complex trade-offs.

**Strengths:**

- The paper considers heuristic design with multiple framework.
- More applicable problems are tested compared with the baselines.

**Weaknesses:**

- The entire framework's success is fundamentally predicated on the quality and diversity of the models the LLM can generate. The search is effectively bounded by the LLM's pre-trained knowledge and potential biases. If the LLM has not been trained on sufficient examples of a certain niche but powerful framework (e.g., specific types of constraint programming or advanced statistical models), it may never propose them, leading to a "rich-get-richer" dynamic where common frameworks are over-explored.
- The paper does not analyze the diversity of the initial framework proposals or the "mutations" suggested in the lower-level search. It's unclear if the LLM is genuinely exploring novel variations or just making superficial syntactic changes to the code. The search is only as good as the operator, and the operator here is a black box.
- The paper validates MOSE's interpretability scores by correlating them with structural/functional complexity proxies (parameter count, permutation entropy). These proxies are not equivalent to interpretability. A model with few parameters can be completely uninterpretable if the parameters lack physical or domain meaning. Conversely, a complex model can be interpretable if its structure is well-understood (e.g., a large but modular compartmental model in epidemiology).
- The core assumption is that an LLM's preference ($p_\mathrm{MOSE}(m_t \succ m_i \mid p)$) is a faithful proxy for that of a human domain expert. This is a very strong and unproven assumption. A small-scale user study with actual ecologists or epidemiologists would be required to truly validate that MOSE captures what experts consider "interpretable." Without this, MOSE is just optimizing for what a specific LLM thinks is interpretable.
- The score is dependent on the initial, fixed reference set $M_{\mathrm{ref}}$. The paper doesn't explore the sensitivity of the results to the composition of this small ($n=3$) reference set.
- The experimental setup involves 20 iterations with a 300-second timeout per model evaluation. This is already a significant time budget. More importantly, each iteration in the lower-level search involves multiple LLM calls: $l=3$ candidate generations, and for each, $q=3$ surrogate estimations. This suggests at least 12 LLM calls per iteration, using a state-of-the-art model (GPT-4o). This approach seems computationally expensive and potentially slow, which may limit its practical application to problems requiring more extensive search or faster turnaround. The paper lacks a discussion on the computational overhead and how it scales.
- The formalism $(m, a)$ and its implementation as a single Python script works well for the self-contained problems chosen. However, real-world mathematical modeling is often a multi-stage pipeline (e.g., data cleaning -> feature engineering -> model selection -> calibration -> post-hoc analysis). It is unclear how the current framework would handle such complex workflows where decisions and trade-offs exist at each stage. The current approach seems to conflate the entire solution process into a single generative step.

**Questions:**

- How does MATHMO guard against the LLM's inherent biases, which might cause it to completely miss entire families of effective models? Is there a risk that the search becomes trapped within the "comfort zone" of the LLM's training data?
- Could you provide more details on the computational cost of a full run of MATHMO? Specifically, what is the wall-clock time and the approximate number of LLM API calls required for a 20-iteration run?
- In Section C.3, you briefly mention a retry mechanism for execution errors. Code generation is notoriously brittle. Could you elaborate on the effectiveness of this self-correction loop? What percentage of generated models fail initially, and what types of errors is the LLM-based correction mechanism successful at fixing?

---

> ### Author Response · Authors · 2025-12-03
> **Response to Reviewer VgEc (Part 1/2)**
>
> *We thank the reviewer for the thoughtful and detailed feedback. We address the main concerns below.*
>
> ---
>
>
> ### [P1] Diversity in model discovery
>
> We appreciate the reviewer’s concern regarding whether the LLM operators genuinely explore semantically meaningful model variations. We respond to this concern in three parts:
> * **Empirical analysis of exploration behavior.** We refer the reviewer to `App C.4` and `App C.5`, where we analyze MATHMO’s exploration at a **semantic** (rather than syntactic) level. These studies show meaningful changes in model performance (across different objectives), indicating they are exploring functionally meaningful model improvements, not syntactic (and thus superficial) changes.
> * **Interpretability of the exploration trajectory.** We note that, while the operator itself is a black box, the entire exploration trajectory is human-interpretable. The sequence of models generated by MATHMO can be inspected, traced, and audited, enabling practitioners to understand the evolution of modeling decisions.
> * To further improve understanding, the revised manuscript now includes representative model–algorithm pairs from the final Pareto frontier for each benchmark, along with annotated Python code and commentary explaining how these discovered models differ from standard baselines.
>
> **[Example annotation]** As an illustrative example, the following summarizes a Pareto-optimal metaheuristic model discovered for the TSP benchmark:
>
> ```
> This model solves the Traveling Salesman Problem (TSP) using a hybrid Ant Colony Optimization (ACO) approach enhanced with 2-opt local search. ACO is a metaheuristic inspired by the behavior of real ants, which use pheromones to find optimal paths. The addition of 2-opt local search improves the refinement of individual ant tours.\n\nSteps of the model:\n1. Pheromone Initialization: A pheromone matrix is initialized with uniform values, representing the initial attractiveness of all city pairs.\n2. Ant Tour Construction: Each ant constructs a tour by iteratively selecting the next city to visit based on a probabilistic rule that combines pheromone intensity (learned desirability) and visibility (inverse of the travel cost).\n3. 2-opt Local Search: After constructing a tour, each ant applies the 2-opt heuristic to refine its solution. This involves reversing segments of the tour to eliminate crossings and reduce the total cost.\n4. Pheromone Update: Pheromones evaporate to reduce their influence over time, and new pheromones are deposited on edges belonging to the ants' tours, proportional to the quality of the tour (inverse of the cost).\n5. Iteration: Steps 2-4 are repeated for a fixed number of iterations. The best tour and cost encountered across all ants and iterations are tracked.\n6. Output: The model returns the best tour found during the optimization process, represented as a permutation of city indices.\n\nThis hybrid approach leverages the global exploration capabilities of ACO and the local optimization power of 2-opt.
>
> MATHMO autonomously discovered this combination, which outperforms other metaheuristics in both solution quality and convergence speed.
> ```
>
> ---
>
> ### [P2] MOSE: expert alignment and sensitivity
>
> The reviewer raises two important concerns: (i) reliability of MOSE as a surrogate of subjective (human expert) preferences, and (ii) sensitivity to the reference set.
>
> **Expert judgement alignment.** We agree that complexity proxies (e.g., parameter count, permutation entropy) are imperfect substitutes for interpretability. For this reason, we conducted a dedicated expert-judgment comparison (detailed in `App A.3`).
> * We collected pairwise interpretability judgments from a domain expert for 25 randomly sampled model pairs.
> * Our analysis found that MOSE achieved a **79.2%** and **76.0%** agreement rate with human experts on Ecology and Epidemiology, respectively. These high agreement rates show that MOSE **tracks meaningful subjective preferences** consistent with expert intuition.
>
> **Sensitivity to reference set.** We also performed a reference-set sensitivity study (`App A.3`). MOSE scores were computed over four independently sampled reference sets, yielding consistently high correlations ($r > 0.93$) between them. This indicates MOSE produces stable interpretability scores that are robust to reference set construction.

---

> > ### Author Response · Authors · 2025-12-03
> > **Response to Reviewer VgEc (Part 2/2)**
> >
> > ### [P3] Extension to multi-stage modeling pipelines
> >
> > We appreciate this conceptual question. MATHMO is designed around a single-step generative operator, but this does not constrain future extensions.
> >
> > While the current instantiation produces a full model/algorithm in a single generative step, the same structure naturally supports multi-step (and more agentic) workflows. Specifically:
> > * LLMs can generate and refine models iteratively within a single exploration step,
> > * Incorporating dedicated data-preprocessing steps, decomposing modeling into sub-tasks, and calling external tools or memory stores.
> >
> > As the first work to explore cross-framework automated mathematical modeling, we intentionally focused on the core challenge of efficient Pareto discovery. We view the extension to multi-stage modeling pipelines as an exciting future direction, and we have clarified this in the revised manuscript.
> >
> > ---
> >
> > ### [P4] Addressing miscellaneous comments
> >
> > **Computational details.** We thank the reviewer for pointing out the need for clarity. In each iteration, only one framework is selected, and a single local exploration step is performed: (1) $l=3$ candidate models are generated; (2) with each receiving $q=3$ surrogate evaluations, resulting in $3\times3$ LLM calls per model generation.
> >
> > **Wall-clock time.** Regarding wall-clock time, each model generation (which can be parallelized) takes roughly 15 seconds. Notably, this is significantly lower than the model solving/fitting process, which can take between $6-20$ minutes, dominating total runtime. While LLM calls add overhead, they remain significantly cheaper than the underlying model evaluations and materially improve sample efficiency.
> >
> > **Self-correction.** Each candidate model may be retried up to three times. Empirically, we observe that (i) failures are more frequent early in a framework’s exploration—typically requiring ∼2 retries—and (ii) failure rates drop substantially in later iterations as the LLM benefits from in-context exemplars in $S_t$. Self-correction is performed by returning the full traceback of any errors or exceptions to the LLM, which effectively resolves most issues (primarily syntax or minor logic errors). We will include a brief quantitative summary of these observations in the revised manuscript.
> >
> >
> > ---
> >
> > *We thank the reviewer again for the thoughtful feedback. We believe the revisions and additional experiments substantially strengthen the paper and address the concerns raised.*

---

### Official Review · Reviewer_N3pu · 2025-11-01

**Soundness:** 3
**Presentation:** 3
**Contribution:** 4
**Rating:** 6
**Confidence:** 4

**Summary:**

This paper introduces MATHMO, a novel system for automated mathematical modeling that leverages Large Language Models (LLMs) within a bi-level adaptive search framework. The work conceptualizes mathematical modeling as a sequential decision-making problem under uncertainty, addressing three key challenges: (1) fundamental modeling uncertainty, (2) balancing multiple conflicting objectives, and (3) incorporating subjective qualities into model evaluation.

The core contribution is a bi-level search architecture where the upper level performs adaptive framework selection using Pareto-Upper Confidence Bound (Pareto-UCB), while the lower level conducts local exploration within selected frameworks using LLM-based surrogates and Bayesian optimization principles. The system incorporates MOSE (Model of Subjective Evaluations), which uses LLMs to approximate human preferences for subjective criteria like interpretability across different modeling frameworks.

The experimental evaluation spans four diverse real-world tasks: Traveling Salesman Problem (TSP), Job Shop Scheduling (JSS), Ecology modeling, and Epidemiology forecasting. Results demonstrate MATHMO's ability to discover diverse Pareto-efficient frontiers of models that effectively balance trade-offs between objectives such as solution quality vs. runtime, and predictive accuracy vs. interpretability.

**Strengths:**

1. Novel problem formulation:  The paper presents a well-motivated and technically sound approach. Its treatment of automated mathematical modeling as a sequential decision-making problem is interesting.
2. Practical impact: Demonstrates feasibility on real-world problems with meaningful trade-offs. Provide inspirations for human-AI collaboration in scientific modeling. The four benchmark problems represent genuine challenges in operations research, ecology, and epidemiology. The ability to discover meaningful trade-offs (e.g., accuracy vs. interpretability) has clear practical value.
3. Technical Quality: The bi-level decomposition is well-motivated and technically sound. The integration of Pareto-UCB for framework selection and LLM-based surrogates for local exploration shows careful consideration of the problem structure.
4. The paper is generally well-written with clear exposition and good organization. The formalization is accessible, and the figures effectively illustrate the key concepts. The related work section provides good context. Areas for improvement include:

**Weaknesses:**

1. Limited Scale and Diversity: While the problems are diverse, the evaluation is limited in scope (4 problems, 20 iterations). More extensive evaluation on larger problem sets would strengthen the claims.
2. Theoretical Analysis: While the approach is intuitive, the paper lacks formal convergence analysis or regret bounds for the proposed adaptive search strategy.The lack of formal convergence guarantees or regret analysis leaves questions about the theoretical properties of the proposed approach.
3.  Baseline Comparisons: While comparisons against MEoH and FunSearch are informative, comparisons against more traditional AutoML baselines (e.g., Bayesian optimization over predefined search spaces) would provide better context for the novelty claims.  The paper relies solely on general LLMs (gpt-4o) with search algorithms and prompt engineering, without exploring the effectiveness of domain-specific fine-tuned models. This approach may miss potential advantages of specialized models in mathematical modeling tasks. The paper does not adequately justify why the bi-level search framework with general LLMs is optimal. Alternative approaches, such as ensemble methods or hybrid search strategies, are not evaluated, leaving questions about design decision rationale and adaptability boundaries.

**Questions:**

1. Scalability: How does the system scale to larger problem instances or longer modeling horizons? The current evaluation is limited to relatively small problems.
2. LLM Prompt Sensitivity: Given the known sensitivity of LLMs to prompt variations, how robust is the system to different prompt designs or LLM model choices?
3. Dynamic Framework Generation: The current approach samples and fixes frameworks at the beginning. Could dynamic framework generation further improve performance?
4. Human-in-the-Loop: How could the system be extended to incorporate real human feedback during the modeling process rather than relying solely on LLM approximations?

---

> ### Author Response · Authors · 2025-12-03
> **Response to Reviewer N3pu (Part 1/2)**
>
> *We thank the reviewer for the thoughtful and constructive feedback. We address the main concerns below.*
>
> ---
>
> ### [P1] Additional evaluations
>
> We agree that broader empirical evaluation strengthens the work. In response, we included two higher-cost, medical risk-scoring modeling tasks, **NHANES** and **SEER**:
> * **NHANES:** is a national health survey containing 86,000 records of demographic, behavioral, clinical, and environmental covariates, where the goal is to predict risk of myocardial infarction.
> * **SEER:** is a population-level cancer registry containing 100,000 patients with incidence, demographic, and survival information, where the goal is to predict risk for breast cancer.
>
> These experiments are introduced as they represent substantially **different domains**, where evaluation is **more expensive** (up to $20$ minutes per model), and involve different **objective trade-offs** (here, discriminative performance of risk-scores AUROC and uncertainty calibration ECE), which are central in clinical modeling.
>
> **Analysis.** We provide detailed results in `App A.1`, but briefly, we observed that across both tasks:
> * **Adaptive search remains effective**: MATHMO discovers diverse Pareto-optimal models under high evaluation cost.
> * **Surrogate-guided selection becomes even more valuable**, reducing wasted evaluations and improving hypervolume.
> * **Exploration across frameworks continues to matter**, with the Pareto frontier drawing from multiple modeling paradigms.
>
> These results demonstrate that the advantages of MATHMO remain viable across multiple domains, and those involving costly real-world evaluations.
>
> ---
>
> ### [P2] Convergence analysis
>
> We appreciate the reviewer’s interest in theoretical grounding. MATHMO’s adaptive bi-level structure generalizes and builds on classical ideas from bandits and multi-objective UCB, but extending formal regret bounds to **LLM-based, cross-framework model generation** is intrinsically challenging because the model space is unbounded and semantically rich.
>
> While theoretical analysis is an important future direction, we highlight empirical investigation demonstrating the adaptive efficiency of upper-level framework selection (in `App C.4`) and lower-level surrogate-guided exploration (in `App C.5`).
>
> ---
>
> ### [P3] Baselines: AutoML, Symbolic Regression, and domain-specific models
>
> We thank the reviewer for these suggestions. Traditional AutoML systems and symbolic regression frameworks focus on exploration within a **single, human-defined model class**, often parameterized by a predefined search space. In contrast, MATHMO performs **cross-framework exploration directly in the model space**, generating and evaluating models across fundamentally different paradigms.
>
> To provide a more complete picture, we include two additional comparisons in `App A.2`:
> * Against highly-optimized, **domain-specific solvers** (Concorde for TSP, OR-Tools for JSSP). These serve as strong gold-standard baselines (i.e., domain-specific fine-tuned models). MATHMO discovers trade-offs competitive with and complementary to solver outputs.
> * Against **AutoML-style symbolic regression** (SINDy for Ecology and Epidemiology). These comparisons illustrate that AutoML approaches, while strong within a single framework, are fundamentally limited in the diversity of models they can produce.
>
> **Analysis.** We defer detailed analysis to `App A.2`. The key takeaways are that (1) competitive performance against expert-designed solvers validates that MATHMO can recover high-performing models and go beyond by further exploring trade-offs across multiple modeling frameworks; and (2) cross-framework search enables MATHMO to discover model classes inaccessible to single-framework AutoML pipelines, yielding systematically stronger and more diverse Pareto sets.

---

> > ### Author Response · Authors · 2025-12-03
> > **Response to Reviewer N3pu (Part 2/2)**
> >
> > ### [P4] Generality across LLMs
> >
> > Thank you for this comment. To assess robustness across LLMs, we repeated experiments using GPT-5 (see `App A.1`). GPT-5 exhibits similar exploration behavior, surrogate performance, and Pareto-front quality.
> >
> > ---
> >
> > ### [P5] Dynamic framework generation, Human-in-the-loop extensions
> >
> > Our current instantiation samples frameworks once at initialization and keeps this set fixed. While this may introduce initialization sensitivity, our method mitigates this via:
> > * **Sequential sampling** at initialization to enforce distinct, non-overlapping frameworks; and
> > * The **Pareto-UCB** policy, which de-emphasizes poor frameworks and reallocates search to promising ones.
> >
> > **Deprioritizing poorer frameworks.** Frameworks that are unproductive are immediately down-weighted by the adaptive exploration process and receive little to no future allocation.
> >
> > **Dynamic framework generation.** We acknowledge (as noted in `L389–391`) that dynamic framework expansion is an important future direction. In preliminary experiments, proposing new frameworks mid-search did not improve performance: for domains like TSP/JSSP, the space of viable paradigm classes is small, and newly proposed frameworks offered little additional diversity and were quickly ignored by Pareto-UCB. We have added this clarification in the revision.
> >
> > **Human-in-the-loop modeling.** MATHMO’s modular structure is well-suited to incorporating expert feedback:
> > * Users could indicate priors to initialize value function estimates for framework selection,
> > * Guide subjective criteria like interpretability (replacing MOSE with real-time feedback), or
> > * Provide rich natural-language feedback to explore promising regions of the model space (which can be naturally incorporated into the MATHMO implementation).
> >
> > We have added a short discussion about this extension in the conclusion.
> >
> > ---
> >
> > *We thank the reviewer again for the thoughtful feedback. We believe the revisions and additional experiments substantially strengthen the paper and address the concerns raised.*

---

### Meta-Review · Area_Chair_wocF · 2025-12-09

**Summary:**

The reviewers had an overall positive reading of the work while identifying weaknesses across multiple dimensions:
- robustness of the results to the choice of LLM or the reference set
- validity of MOSE to capture model interpretability
- limited scale of the datasets for evaluation
- comparison with other baselines (AutoML) and other expert models

**Reviewer Concerns:**

I found the authors' response to be clear and convincing. While some more minor points remain, the major points cited above have been addressed, with another LLM being tested, a small user study being performed on MOSE, and more baselines and tasks being added.

**Reviewer Scores:**

The paper received scores 6, 6, 6, and 4, and were hence leaning towards acceptance. I believe the careful response would have pushed the scores up and confirmed the acceptance.

---

### Decision · Program_Chairs · 2026-01-26

Accept (Poster)